# Multiple ciliary localization signals control INPP5E ciliary targeting

**Dario Cilleros-Rodriguez**[1,2,3,4†], **Raquel Martin-Morales**[1,2,3,4†], **Pablo Barbeito**[1,2,3,4], **Abhijit Deb Roy**[5], **Abdelhalim Loukil**[6], **Belen Sierra-Rodero**[1,2,3,4], **Gonzalo Herranz**[1,2], **Olatz Pampliega**[7], **Modesto Redrejo-Rodriguez**[1,2], **Sarah C Goetz**[6], **Manuel Izquierdo**[1,2], **Takanari Inoue**[5], **Francesc R Garcia-Gonzalo**[1,2,3,4*]

[1]Departamento de Bioquímica, Facultad de Medicina, Universidad Autónoma de Madrid (UAM), Madrid, Spain; [2]Instituto de Investigaciones Biomédicas "Alberto Sols" (IIBM), Consejo Superior de Investigaciones Científicas (CSIC)-UAM, Madrid, Spain; [3]Instituto de Investigación del Hospital Universitario de La Paz (IdiPAZ), Madrid, Spain; [4]CIBER de Enfermedades Raras (CIBERER), Instituto de Salud Carlos III (ISCIII), Madrid, Spain; [5]Department of Cell Biology, Center for Cell Dynamics, Johns Hopkins University School of Medicine, Baltimore, United States; [6]Department of Pharmacology and Cancer Biology, Duke University School of Medicine, Durham, United States; [7]Department of Neurosciences, University of the Basque Country, Achucarro Basque Center for Neuroscience-UPV/EHU, Leioa, Spain

**Abstract** Primary cilia are sensory membrane protrusions whose dysfunction causes ciliopathies. INPP5E is a ciliary phosphoinositide phosphatase mutated in ciliopathies like Joubert syndrome. INPP5E regulates numerous ciliary functions, but how it accumulates in cilia remains poorly understood. Herein, we show INPP5E ciliary targeting requires its folded catalytic domain and is controlled by four conserved ciliary localization signals (CLSs): LLxPIR motif (CLS1), W383 (CLS2), FDRxLYL motif (CLS3) and CaaX box (CLS4). We answer two long-standing questions in the field. First, partial CLS1-CLS4 redundancy explains why CLS4 is dispensable for ciliary targeting. Second, the essential need for CLS2 clarifies why CLS3-CLS4 are together insufficient for ciliary accumulation. Furthermore, we reveal that some Joubert syndrome mutations perturb INPP5E ciliary targeting, and clarify how each CLS works: (i) CLS4 recruits PDE6D, RPGR and ARL13B, (ii) CLS2-CLS3 regulate association to TULP3, ARL13B, and CEP164, and (iii) CLS1 and CLS4 cooperate in ATG16L1 binding. Altogether, we shed light on the mechanisms of INPP5E ciliary targeting, revealing a complexity without known parallels among ciliary cargoes.

## Editor's evaluation

This manuscript is of interest to readers in the field of cilia biology and ciliopathies. The authors address the molecular mechanisms by which INPP5E, a ciliary phosphoinositide phosphatase mutated in multiple ciliopathies, is targeted to the primary cilium of cultured mammalian cells. By combining cell-based analysis of various INPP5E mutant constructs with biochemical assays and structure prediction, the authors show that ciliary targeting of INPP5E requires its folded catalytic domain and is controlled by four ciliary localization signals. The work clarifies and extends previous work in the field and reveals a complex ciliary targeting mechanism unparalleled among other known ciliary cargoes. The claims are generally well supported by the data.

*For correspondence:
francesc.garcia@uam.es

†These authors contributed equally to this work

## Introduction

Primary cilia are solitary membrane protrusions acting as cellular antennae. They emanate from the basal body, a specialized mother centriole, and consist of a microtubule shaft, or axoneme, surrounded by the ciliary membrane, which is topologically continuous with, but compositionally distinct from, the plasma membrane (PM). For cilia to perform their signaling functions, they must accumulate specific receptors and signal transducers. For this to happen, these proteins must first reach the ciliary base, from where they can enter cilia by crossing the transition zone (TZ), the border region separating the ciliary compartment from the rest of the cell. If they can make it inside cilia, the ciliary levels of these proteins will then depend on the balance between ciliary entry and exit rates, a balance that can shift over time. Ciliary entry and exit rates in turn depend on how proteins interact with TZ components, and on whether they associate with specialized ciliary trafficking machinery, such as intraflagellar transport (IFT) trains, microtubule motor-driven multiprotein assemblies whose components, like the IFT-B, IFT-A, and BBSome complexes, selectively bind ciliary cargoes to mediate their transport into or out of cilia (*Nachury and Mick, 2019*; *Reiter and Leroux, 2017*; *Garcia-Gonzalo and Reiter, 2017*).

Ciliary malfunction causes ciliopathies, a diverse group of human diseases, many of which are rare autosomal recessive syndromes. One such disease is Joubert syndrome (JBTS), affecting ≈ 1 in 100,000 people worldwide and whose pathognomonic signature is the molar tooth sign (MTS), a cerebellar and midbrain malformation observable by magnetic resonance imaging (MRI). JBTS patients may also present with mild to severe intellectual disability, hypotonia, ataxia, oculomotor apraxia, apnea/hyperpnea, polydactyly, kidney cysts, and retinal dystrophy. Genetically, JBTS can be caused by mutations in over 30 different genes, all involved in ciliary function (*Reiter and Leroux, 2017*; *Braun and Hildebrandt, 2017*; *Bachmann-Gagescu et al., 2020*; *Bachmann-Gagescu et al., 2015*).

More specifically, JBTS-causative genes regulate a ciliary signaling network, one of whose main nodes is the ciliary phosphoinositide phosphatase INPP5E (Inositol polyphosphate-5-phosphatase E, formerly known as Pharbin, or as type IV 72 kDa 5-phosphatase) (*Jacoby et al., 2009*; *Bielas et al., 2009*; *Asano et al., 1999*; *Kisseleva et al., 2000*; *Kong et al., 2000*). Evidence for the central role of INPP5E in JBTS includes: (i) INPP5E is one of the most commonly mutated JBTS genes (*Bachmann-Gagescu et al., 2015*); (ii) mouse models of INPP5E loss of function recapitulate key features of human JBTS, including the axon tract defects leading to MTS in humans (*Jacoby et al., 2009*; *Guo et al., 2019*); (iii) most other JBTS genes encode proteins required for INPP5E ciliary targeting (*Garcia-Gonzalo and Reiter, 2017*; *Bachmann-Gagescu et al., 2020*; *Garcia-Gonzalo et al., 2011*; *Roberson et al., 2015*; *Slaats et al., 2016*; *Alkanderi et al., 2018*; *Thomas et al., 2014*; *Humbert et al., 2012*; *Ning et al., 2021*; *Dowdle et al., 2011*); and (iv) the few JBTS genes that may not be needed for INPP5E ciliary targeting regulate the same pathways as INPP5E, like Hedgehog (Hh) signaling or ciliary stability (*Latour et al., 2020*; *Dafinger et al., 2011*; *De Mori et al., 2017*; *Van De Weghe et al., 2017*; *Frikstad et al., 2019*; *Ki et al., 2020*; *Lee et al., 2012*). Through its effects on Hh and phosphatidylinositol 3-kinase (PI3K) signaling, INPP5E also promotes tumor progression in medulloblastoma (*Conduit et al., 2017*).

Almost all JBTS-causing INPP5E mutations are missense mutations affecting its catalytic domain (*Bachmann-Gagescu et al., 2015*; *Travaglini et al., 2013*; *de Goede et al., 2016*). In all cases tested, these mutations impaired 5-phosphatase activity toward one or both of its main substrates, PI(4,5)$P_2$ and PIP$_3$ (*Bielas et al., 2009*; *de Goede et al., 2016*). INPP5E mutations can also cause other ciliopathies, including retinitis pigmentosa (RP), Leber congenital amaurosis (LCA), and MORM syndrome (mental retardation, obesity, retinal dystrophy, and micropenis in males). Unlike JBTS, RP and LCA, the MORM mutation does not affect the catalytic domain, instead removing INPP5E's C-terminal CaaX box, whose farnesylation tethers INPP5E to membrane (*Jacoby et al., 2009*; *de Goede et al., 2016*; *Xu et al., 2015*; *Wang et al., 2013*). Besides its catalytic domain and C-terminus, INPP5E also contains a proline-rich N-terminal region, where no ciliopathy mutations have been reported (*Jacoby et al., 2009*; *Bielas et al., 2009*; *Asano et al., 1999*; *Kisseleva et al., 2000*; *Kong et al., 2000*; *de Goede et al., 2016*).

INPP5E plays multiple important roles at the cilium. Among others, these roles include regulation of: (i) ciliary phosphoinositide levels, (ii) ciliary protein composition, (iii) ciliary Hedgehog and PI3K signaling, (iv) ciliary ectovesicle release, (v) ciliary stability, and (vi) ciliogenesis (*Jacoby et al., 2009*; *Bielas et al., 2009*; *Guo et al., 2019*; *Garcia-Gonzalo et al., 2015*; *Chávez et al., 2015*; *Badgandi et al., 2017*; *Dyson et al., 2017*; *Phua et al., 2017*; *Wang et al., 2011*; *Hakim et al., 2016*; *Xu et al.,*

*2016*; *Xu et al., 2017*; *Constable et al., 2020*; *Hasenpusch-Theil et al., 2020*; *Sharif et al., 2021*; *Ukhanov et al., 2022*; *Yue et al., 2021*). Although most of its functions are ciliary, INPP5E also plays extraciliary roles, like promoting autophagosome-lysosome fusion during autophagy (*Hasegawa et al., 2016*).

Two motifs in INPP5E protein sequence are known to affect its ciliary targeting: the FDRxLYL motif (aa 609–615) and the CaaX box (aa 641–644), both located in the C-terminal region after the phosphatase domain (aa 297–599) (*Jacoby et al., 2009*; *Thomas et al., 2014*; *Humbert et al., 2012*; *Kösling et al., 2018*; *Fujisawa et al., 2021*; *Qiu et al., 2021*). The CaaX box is not essential for INPP5E ciliary localization, but CaaX box mutants show reduced ciliary targeting, while increasing its proportion at the ciliary base and elsewhere (*Jacoby et al., 2009*; *Thomas et al., 2014*; *Kösling et al., 2018*). CaaX box farnesylation allows INPP5E to bind phosphodiesterase 6 subunit delta (PDE6D), a prenyl-binding protein that extracts INPP5E from the ciliary base membrane and ferries it across the transition zone. Once inside the cilium, the active form of the monomeric G-protein ADP ribosylation factor (ARF)-like 3 (ARL3) induces dissociation of the PDE6D-INPP5E complex, thus releasing the farnesyl group for insertion into the ciliary membrane. For this to happen, ARL3 must first be activated by a guanine nucleotide exchange factor (GEF) complex consisting of ARF-like 13B (ARL13B), an atypical small G-protein, and its cofactor Binder of ARL2 (BART) (*Thomas et al., 2014*; *Humbert et al., 2012*; *Fujisawa et al., 2021*; *Gotthardt et al., 2015*; *Fansa et al., 2016*; *Ivanova et al., 2017*; *ElMaghloob et al., 2021*).

Intriguingly, despite INPP5E farnesylation not being essential for ciliary targeting, the latter is completely dependent on PDE6D, ARL3, and ARL13B, all of them also JBTS-causative genes (*Bachmann-Gagescu et al., 2020*; *Alkanderi et al., 2018*; *Thomas et al., 2014*; *Humbert et al., 2012*). Although the reason for this apparent discrepancy is unknown, part of the answer might relate to the retinitis pigmentosa GTPase regulator (RPGR), an RP-associated protein required for INPP5E ciliary targeting, and whose own ciliary localization depends on its geranylgeranylated CaaX box binding to PDE6D (*Zhang et al., 2019*; *Rao et al., 2016*; *Dutta and Seo, 2016*; *Lee and Seo, 2015*; *Fansa et al., 2015*; *Wätzlich et al., 2013*; *Vössing et al., 2021*).

Regarding ARL13B, its direct interaction with INPP5E probably explains its strong requirement for INPP5E ciliary targeting (*Humbert et al., 2012*; *Fujisawa et al., 2021*; *Qiu et al., 2021*). This interaction is mediated by the FDRxLYL motif, which, unlike the CaaX box, is absolutely required for INPP5E ciliary localization (*Humbert et al., 2012*). Ciliary localization of ARL13B is in turn dependent on Tubby-like protein 3 (TULP3), a phosphoinositide-binding adaptor that links ciliary membrane cargoes to IFT trains (*Badgandi et al., 2017*; *Han et al., 2019*; *Mukhopadhyay et al., 2010*; *Mukhopadhyay et al., 2013*; *Palicharla et al., 2021*). This probably explains why TULP3 is also required for INPP5E ciliary targeting, although a more direct connection between TULP3 and INPP5E might also exist (*Han et al., 2019*; *Palicharla et al., 2021*). Other proteins needed for INPP5E ciliary targeting include the centrosomal protein of 164 kDa (CEP164), which is involved in INPP5E recruitment to the ciliary base (*Humbert et al., 2012*; *Xu et al., 2016*; *Cajanek and Nigg, 2014*; *Schmidt et al., 2012*; *Graser et al., 2007*), and the autophagy-related protein 16-like 1 (ATG16L1), also implicated in ciliogenesis and ciliary trafficking (*Boukhalfa et al., 2021*; *Pampliega et al., 2013*).

Although the C-terminal region of INPP5E contains both FDRxLYL motif and CaaX box, this region alone is not sufficient to target INPP5E to cilia. In contrast, a fragment containing both phosphatase domain and C-terminal region suffices for ciliary accumulation (*Humbert et al., 2012*). This indicates that something in or near the catalytic domain is also essential for ciliary targeting, but the reasons for this requirement are unknown.

Herein, we start by elucidating why the catalytic domain is required for ciliary targeting. There are two reasons for this: (i) the FDRxLYL motif is part of the catalytic domain's globular fold, even though the motif is outside the conserved domain as defined by primary sequence analysis; and (ii) a key catalytic domain residue, W383, which is physically adjacent to the FDRxLYL motif in the domain's crystal structure, is also specifically and strongly required for ciliary accumulation. We then resolve another lingering question in the field: why the CaaX box is dispensable for INPP5E ciliary localization. We show that the CaaX box is partially redundant with the LLxPIR motif, located near the end of the N-terminal region. Thus, while single deletion of CaaX box or LLxPIR moderately reduces ciliary targeting, simultaneous deletion of both completely abolishes it. Therefore, we reveal that INPP5E ciliary accumulation depends on the interplay between four different ciliary localization signals (CLS1-4), the first

two of which we newly identify here: the LLxPIR motif (CLS1), W383 (CLS2), the FDRxLYL motif (CLS3) and the CaaX box (CLS4). In the second half of this work, we systematically examine how each of these CLSs affects INPP5E interactions with proteins required for its ciliary targeting. Through this approach, we find that CLS2 and CLS3 function by promoting interaction with TULP3 and ARL13B, while also antagonizing CEP164 binding. On the other hand, CLS4 is needed for association to PDE6D, RPGR and, in cooperation with CLS1, to ATG16L1. Altogether, our data reveal an unprecedented degree of complexity in the ciliary targeting mechanisms of INPP5E, as compared to other known ciliary cargoes, for which a single or at most two CLSs suffice to explain ciliary accumulation (*Nachury and Mick, 2019*; *Garcia-Gonzalo and Reiter, 2012*; *Barbeito and Garcia-Gonzalo, 2021*; *Mukhopadhyay et al., 2017*; *McIntyre et al., 2016*).

## Results

### INPP5E catalytic domain encompasses the FDRxLYL motif and is required for ciliary targeting

Ciliary accumulation of human INPP5E requires the conserved FDRxLYL motif (aa 609–615) (*Humbert et al., 2012*). This motif lies downstream of INPP5E's highly conserved phosphatase domain, as defined by the InterPro protein signature database (InterPro domain IPR000300, aa 297–599) (*Humbert et al., 2012*; *Blum et al., 2021*). However, INPP5E crystal structure reveals a more extensive globular phosphatase domain, spanning residues 282–623, on whose surface the FDRxLYL motif folds (PDB ID: 2xsw) (*Figure 1a*; *Tresaugues et al., 2010*). Indeed, the crystallographic data show that the FDRxLYL motif, and the alpha-helix it is nested in, interact with several other catalytic domain residues (such interactions include D610-L362, Y614-P358, R620-E347, and R621-E354). Recently, a 3D model of full-length INPP5E was generated using AlphaFold, a remarkably accurate machine learning algorithm for protein structure prediction (*Jumper et al., 2021*). The AlphaFold model closely matches the crystal structure, including the FDRxLYL motif's structure and location. Thus, based on the structural data, we conclude that INPP5E's catalytic domain spans residues 282–623, and therefore encompasses the FDRxLYL motif, which lies on the domain's surface. Accordingly, from now on, when we speak of the catalytic or phosphatase domain, we will be referring to the globular domain spanning residues 282–623, unless we specify otherwise.

To clarify how INPP5E's phosphatase domain controls ciliary targeting, we generated a series of INPP5E deletion mutants lacking different portions of the N or C-terminal regions, or lacking only the InterPro-defined catalytic domain (*Figure 1b*). Interestingly, INPP5E ciliary accumulation closely correlated with the presence of an intact structurally-defined catalytic domain (aa 282–623) (*Figure 1b–c*). Deletion of the InterPro-defined catalytic domain (Δ297–599) completely abolished ciliary targeting in EGFP-INPP5E-transfected hTERT-RPE1 cells (*Figure 1c*). N-terminal deletions not affecting the phosphatase domain did not affect ciliary localization, as was the case for constructs 100–644, 200–644, 251–644, and 274–644 (*Figure 1c*). In contrast, 288–644 displayed strongly reduced ciliary targeting, and 351–644, 451–644 and 551–644 completely failed to localize to cilia (*Figure 1c*). C-terminal deletions showed a similar pattern. As previously reported, the MORM mutant (1-626) was found inside cilia and at the ciliary base, and the same was true for 1–623 (*Figure 1c*; *Jacoby et al., 2009*). The 1–621 mutant was only occasionally ciliary, at low levels near the ciliary base (*Figure 1c*). Further deletions from the C-terminus resulted in complete loss of ciliary targeting, as was the case for 1–618, 1–616, 1–608, and 1–283 (*Figure 1c*). Hence, integrity of the structurally defined catalytic domain is essential for INPP5E ciliary accumulation.

At least partly, this lack of ciliary targeting could be due to protein instability of these mutants. To assess this, we used western blot to measure the expression levels of many of these mutants in transfected HEK293T cells. Given the nature of this expression system, we expect any changes in protein levels between constructs to be likely due to changes in protein stability, at least when comparing proteins of similar sizes (if not, then slower translation rates of longer mRNAs need to be accounted for). Other factors affecting protein levels in other contexts are unlikely to play important roles in this system, including changes in transfection efficiency (always very high in HEK293T), transcriptional regulation (expression of all constructs driven by constitutive and strong cytomegalovirus promoter), or post-transcriptional mRNA regulation (no anticipated differences between constructs).

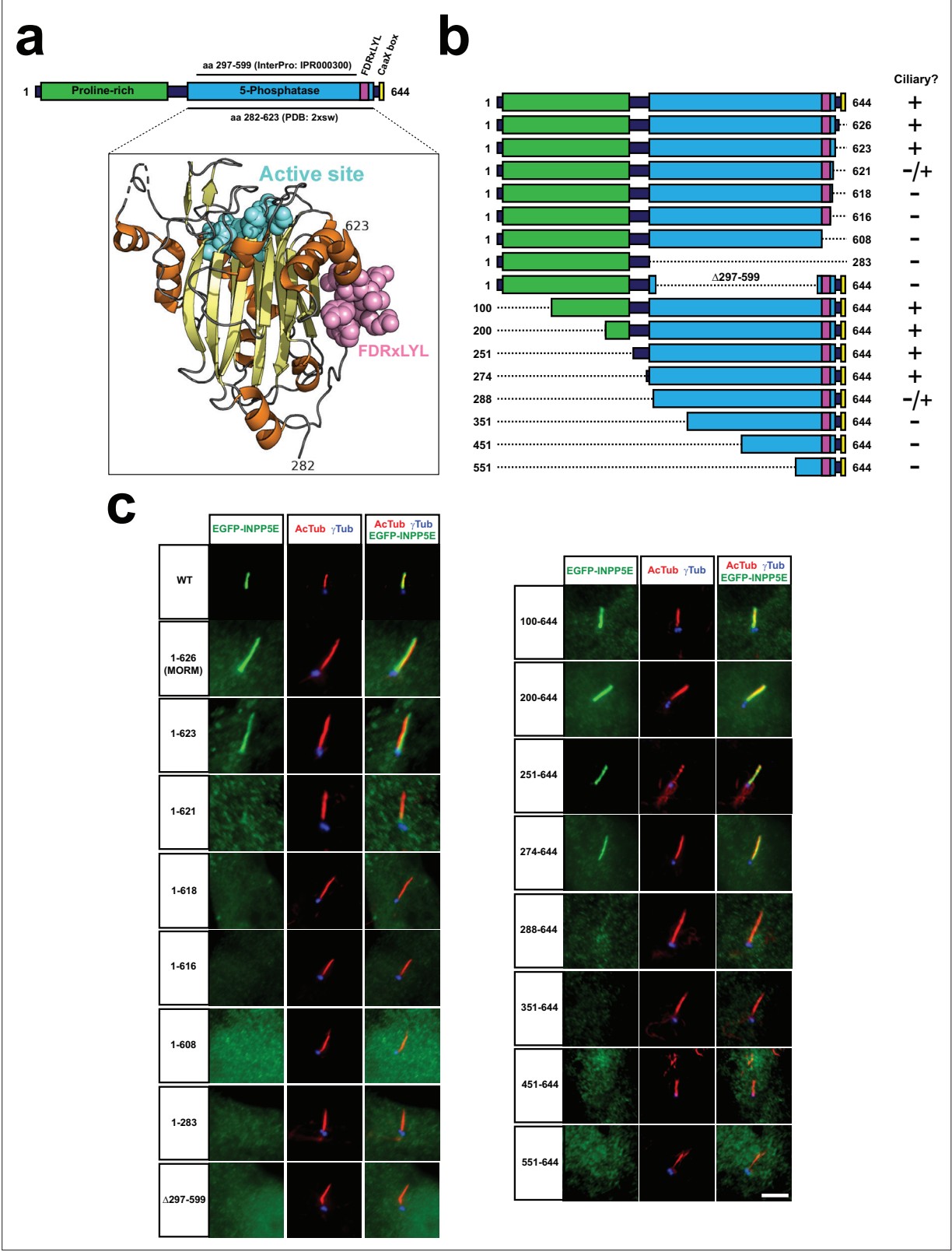

**Figure 1.** INPP5E catalytic domain encompasses FDRxLYL motif and is required for ciliary targeting. (**a**) Top diagram represents full length human INPP5E protein sequence (aa 1–644). Depicted are the proline-rich region (aa 10–242, Uniprot), the previously reported ciliary localization signal (FDRxLYL, aa 609–615 *Humbert et al., 2012*), and the CaaX box driving farnesylation (aa 641–644). Also shown is the inositol polyphosphate 5-phosphatase catalytic domain, whose most conserved core corresponds to InterPro domain IPR000300 (aa 297–599), but which actually spans aa

*Figure 1 continued on next page*

*Figure 1 continued*

282–623, as revealed by its crystal structure, available at the Protein Data Bank (PDB) and displayed below (PDB ID: 2xsw). Notice how FDRxLYL residues (in magenta above and below) are part of the catalytic domain, on whose surface they fold. The 3D structure also shows active site residues in cyan, alpha-helices in orange, and beta-strands in yellow (including the beta-sandwich at the domain's core, and a small beta-hairpin near the active site). (**b**) Schematic representation of full length human INPP5E (1-644) and its deletion mutants used in (**c**), indicating on the right which ones localize to cilia. (**c**) Immunofluorescence images of cilia from hTERT-RPE1 cells transfected with the indicated EGFP-INPP5E constructs. Cells were stained with antibodies against acetylated α-tubulin (AcTub), γ-tubulin (γTub) and EGFP to detect the fusion proteins. Images are representative of at least two independent experiments per construct, with >30 transfected-cell cilia visualized per construct and experiment. Scale bar, 5 µm.

The online version of this article includes the following source data and figure supplement(s) for figure 1:

**Figure supplement 1.** Expression levels of EGFP-INPP5E constructs from *Figure 1*.

**Figure supplement 1—source data 1.** Uncropped immunoblots from *Figure 1—figure supplement 1*.

**Figure supplement 1—source data 2.** Uncropped immunoblot from *Figure 1—figure supplement 1* (WB: EGFP).

**Figure supplement 1—source data 3.** Uncropped immunoblot from *Figure 1—figure supplement 1* (WB: Tubulin).

**Figure supplement 1—source data 4.** Uncropped immunoblot from *Figure 1—figure supplement 1* (WB: EGFP).

**Figure supplement 1—source data 5.** Uncropped immunoblot from *Figure 1—figure supplement 1* (WB: Tubulin).

When we did these experiments, we observed a clear effect of the catalytic domain (aa 282–623) on EGFP-INPP5E protein levels (*Figure 1—figure supplement 1*). This was very clear, for instance, when comparing 1–626 to 1–618, and also 274–644 to 288–644. In both cases, as catalytic domain residues where removed in the second mutant of each pair, protein levels decreased sharply, by at least 4-fold. On the other hand, virtually complete removal of the catalytic domain in Δ297–599 only reduced expression moderately, about 2-fold. Yet ciliary targeting of this mutant was completely abolished, not just reduced by half (*Figure 1c*). Altogether, these data suggest that the catalytic domain, in addition to affecting protein stability, has a specific role in promoting INPP5E ciliary accumulation.

## W383 and FDRxLYL function as specific CLSs on the catalytic domain surface

We first confirmed that, as previously reported, the FDRxLYL motif is essential for INPP5E ciliary targeting (*Figure 2a*; *Humbert et al., 2012*). Mutation to alanines of both the FDR (aa 609–611) and LYL (aa 613–615) triplets completely abolished ciliary localization (*Figure 2a*). To assess the relative importance of each residue within the FDRxLYL motif, we also made the individual alanine mutants (F609A, D610A, R611A, E612A, L613A, Y614A, L615A). Surprisingly, all of them still localized to cilia, indicating redundancy within the FDR and LYL triplets (*Figure 2—figure supplement 1*).

Given that the FDRxLYL motif lies on the catalytic domain surface (*Figure 1a*), and that catalytic domain integrity is essential for ciliary targeting (*Figure 1b–c*), we hypothesized that INPP5E ciliary targeting relies on a catalytic domain surface including not only the FDRxLYL residues, but also other residues whose proximity to FDRxLYL is dependent upon a folded domain. To test this, we examined the 3D structure of INPP5E's catalytic domain (PDB ID: 2XSW) in order to identify candidate residues for this putative surface. We did this by selecting residues meeting all or most of the following criteria: (i) located near FDRxLYL motif, on same side of domain; (ii) high exposure to solvent; and (iii) highly conserved in vertebrate INPP5E orthologs. Out of this analysis, we selected sixteen candidate residues: R345, R346, E347, W348, E349, Q353, E354, Y360, V361, R378, R379, D380, I382, W383, F384, E387. Alanine mutation of most of these residues did not affect ciliary targeting of EGFP-INPP5E in hTERT-RPE1 cells (*Figure 2b–c*). Exceptions were W348A, Y360A, W383A, and R378A+R379A, all of which completely failed to accumulate in cilia (*Figure 2c–d*). The R378 and R379 arginines are redundant, as the double but not single mutations impeded cilia localization (*Figure 2c–d*).

Mistargeting of these four mutants could be due to loss of catalytic domain integrity, which is critical for ciliary targeting (*Figure 1*). If so, then phosphatase activity would also be disrupted in these mutants. To test this, we immunoprecipitated these EGFP-INPP5E mutants from transfected HEK293T lysates and measured their $PI(4,5)P_2$ 5-phosphatase activity in the immunoprecipitates (IPs). EGFP-INPP5E wild type was used as positive control, whereas negative controls included: (i) a reaction with substrate but no enzyme, to assess the rate of basal $PI(4,5)P_2$ dephosphorylation; and (ii) the catalytically inactive D477N INPP5E mutant, lacking a critical active site aspartate but normally

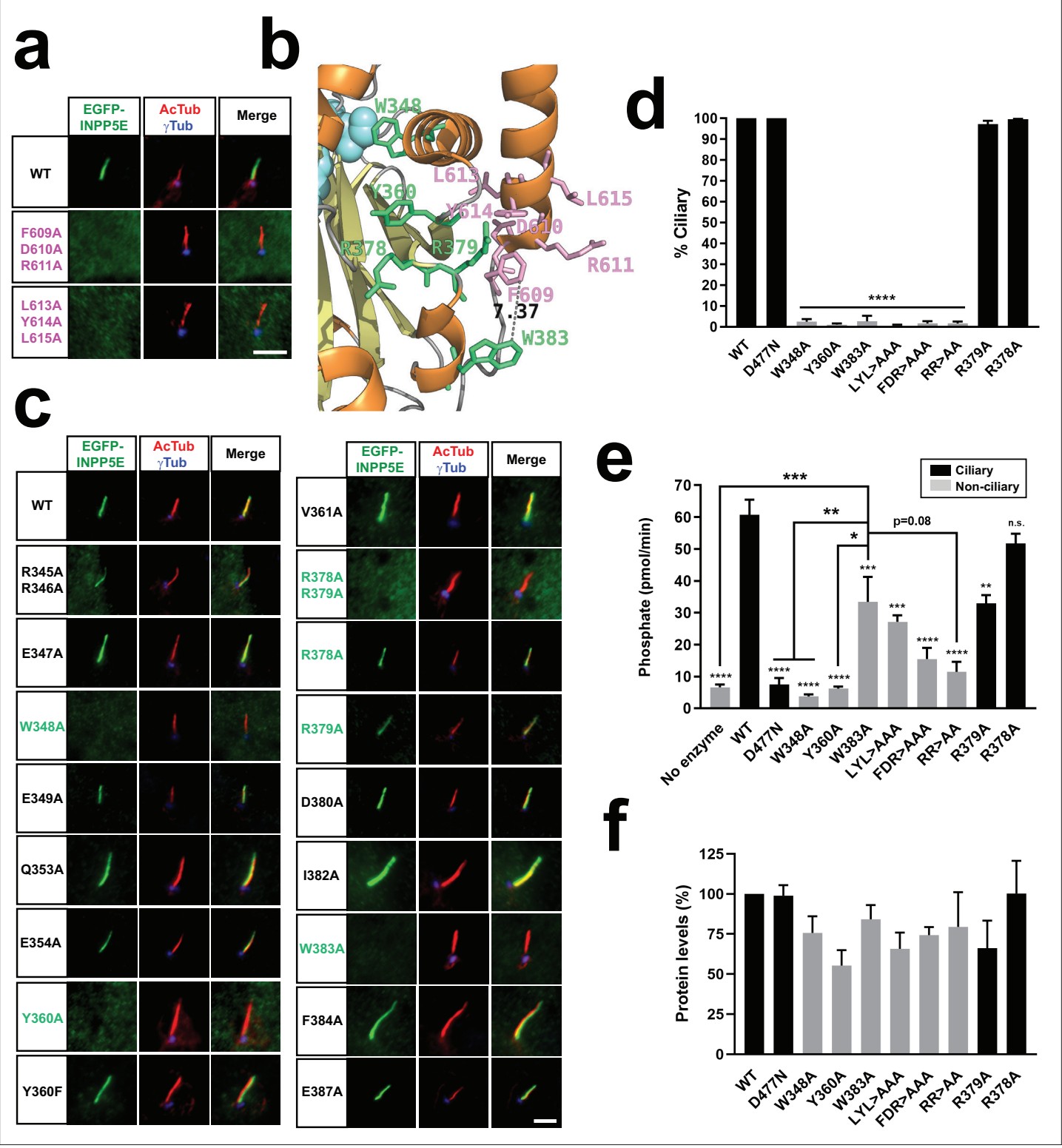

**Figure 2.** W383 and FDRxLYL motifs act as CLSs on the catalytic domain surface. (**a**) Cilia localization of the indicated FDRxLYL mutants of EGFP-INPP5E was analyzed in hTERT-RPE1 cells as in *Figure 1*. Scale bars, 5 μm. (**b**) Magnification from INPP5E structure (PDB ID: 2xsw) showing the FDRxLYL motif residues (pink) and adjacent catalytic domain residues shown here to affect ciliary targeting (green). Distance between W383 and F609 is indicated in angstroms. Beta-sheets and alpha-helices are shown as yellow and orange ribbons, respectively. Notice active site region on top left (cyan). (**c**) Cilia localization of the indicated EGFP-INPP5E constructs was analyzed as in (**a**). In both cases, images are representative of at least two independent experiments per construct, with >30 transfected-cell cilia visualized per construct and experiment. Scale bars, 5 μm. (**d**) Percentage of positive cilia

*Figure 2 continued on next page*

*Figure 2 continued*

was quantitated for each of the indicated constructs. Data are mean ± SEM of n=3 independent experiments. Data were analyzed by one-way ANOVA followed by Tukey's multiple comparisons tests. Significance relative to WT is shown as p<0.0001(****). LYL >AAA: L613A+Y614A+L615 A; FDR >AAA: F609A+D610A+R611 A; RR >AA: R378A+R379 A. (**e**) 5-phosphatase activity, expressed as picomoles of released inorganic phosphate per minute, was measured, using PI(4,5)P2 as substrate, in immunoprecipitates of HEK293T cells transfected with the indicated EGFP-INPP5E variants. Cilia-localized constructs shown as black columns, non-ciliary as grey. Data are mean ± SEM of n=9,9,5,4,3,5,3,3,2,3,3 independent experiments (from left to right). Data were analyzed by one-way ANOVA followed by Tukey's multiple comparisons tests. Significance relative to WT is shown as small asterisks directly above each bar. Significance relative to W383A is shown as bigger asterisks as indicated. In all cases, significance is represented as: p<0.05(*), p<0.01(**), p<0.001(***), p<0.0001(****), or n.s. (not significant). (**f**) Protein levels in the immunoprecipitates used for the activity assays in (**e**). Western blot bands were quantitated and plotted as percentage of WT. Data are mean ± SEM of n=8,5,3,3,5,2,3,3,2,3 independent experiments (from left to right). One-way ANOVA revealed no significant differences.

The online version of this article includes the following source data and figure supplement(s) for figure 2:

**Source data 1.** Source data for *Figure 2d*.

**Source data 2.** Source data for *Figure 2e*.

**Source data 3.** Source data for *Figure 2f*.

**Figure supplement 1.** FDRELYL motif residues are not individually required for INPP5E cilia localization.

**Figure supplement 2.** Expression levels of EGFP-INPP5E constructs from *Figure 2*.

**Figure supplement 2—source data 1.** Uncropped immunoblots from *Figure 2—figure supplement 2*.

**Figure supplement 2—source data 2.** Uncropped immunoblot from *Figure 2—figure supplement 2* (WB: EGFP).

**Figure supplement 2—source data 3.** Uncropped immunoblot from *Figure 2—figure supplement 2* (WB: Tubulin).

**Figure supplement 3.** Stability of W383A and FDRxLYL mutants.

**Figure supplement 3—source data 1.** Uncropped immunoblots from *Figure 2—figure supplement 3a*.

**Figure supplement 3—source data 2.** Uncropped immunoblots from *Figure 2—figure supplement 2A* (WB: EGFP for WT and W383A).

**Figure supplement 3—source data 3.** Uncropped immunoblot from *Figure 2—figure supplement 2A* (WB: EGFP for FDR >AAA and LYL >AAA).

**Figure supplement 3—source data 4.** Uncropped immunoblot from *Figure 2—figure supplement 2A* (WB: Tubulin for WT and W383A).

**Figure supplement 3—source data 5.** Uncropped immunoblot from *Figure 2—figure supplement 2A* (WB: Tubulin for FDR >AAA and LYL >AAA).

**Figure supplement 4.** Effect of W383 mutations on ciliary targeting and activity.

**Figure supplement 4—source data 1.** Uncropped immunoblot from *Figure 2—figure supplement 4c*.

**Figure supplement 4—source data 2.** Uncropped immunoblot from *Figure 2—figure supplement 4c* (WB: EGFP).

localizing to cilia (*Bielas et al., 2009*; *Kong et al., 2000*; *Garcia-Gonzalo et al., 2015*; *Kong et al., 2006*; *Whisstock et al., 2000*).

Compared to negative controls, phosphate release by WT was about 12-fold faster, a very significant difference (*Figure 2e*). W348A and Y360A were completely inactive, and R378A+R379 A nearly so (*Figure 2e*). Hence, lack of ciliary targeting in these mutants may be non-specific. Unlike R378A+R379 A, the cilia-localized R378A and R379A single mutants retained activity, either fully (R378A) or partially (R379A). Like R379A, W383A was half as active as WT, with a very significant ≈6-fold increase over negative controls (*Figure 2e*). The same was true for the LYL triplet mutant, whereas the FDR mutant was somewhat less active, with a ≈3-fold increase over controls (*Figure 2a–e*). Quantitation by anti-EGFP immunoblot of the protein levels of each mutant in the IPs used to measure activities revealed no significant differences for any of the mutants (*Figure 2f*). We also analyzed expression levels of these mutants at 48 hr post-transfection by Western blot of lysates in a separate HEK293T cell experiment, in which protein levels of the W383A, LYL and FDR mutants were ≈75%, 60%, and 50% of WT, respectively. The lowest expression was seen for W348A, Y360A, R378A+R379 A, and R379A, all between 20 and40% of WT (*Figure 2—figure supplement 2*).

All of this suggests that W383 and FDRxLYL function as bona fide CLSs, as their strict requirement for ciliary targeting cannot fully be accounted for by their moderate effects on enzyme activity or protein levels. Accordingly, R379A, a mutation immediately adjacent to FDRxLYL and W383, displayed similar moderate effects on activity and protein levels, but had no effect on ciliary localization (*Figure 2b–f*).

Still, the partial activity loss in W383A and FDRxLYL mutants suggests that catalytic domain integrity may be partially lost as well. To examine this further, we assessed the half-lives of these mutants in HEK293T cells using the protein translation inhibitor cycloheximide (*Figure 2—figure supplement 3*). After 5, 10 or 24 hr in cycloheximide, EGFP-INPP5E(WT) levels were virtually unaffected, indicating

a long half-life of several days. For W383A and the triplet FDR and LYL mutants, initial levels, at 24 h post-transfection, were not significantly different from WT (*Figure 2—figure supplement 3a*-**b**). From those initial levels (100%), W383A fell to 60% in the first 10 hr, but then remained at 60% by 24 hr, rather than diminishing further (*Figure 2—figure supplement 3c*). Similar results were seen for the FDR mutant, whereas LYL's curve ran closer to WT's (*Figure 2—figure supplement 3c*).

Thus, W383A showed a biphasic kinetics, whose most parsimonious explanation appears to be the existence of two protein populations: ≈40% of W383A would be unstable with a half-life of ≈5 hr, whereas the remaining ≈60% would be stable, with a half-life of days, like WT (*Figure 2—figure supplement 3c*). Presumably, the unstable form would not be properly folded and would be inactive, whereas the stable form would be folded and active, which would explain why W383A reduces activity ≈2-fold (*Figure 2e*). Although this model remains speculative, the fact that ≈50–60% of W383A is stable and enzymatically active supports the idea that W383 is a bona fide CLS, since loss of ciliary targeting in W383A is virtually complete, and much stronger than a twofold reduction (*Figure 2c–d*). Similar points support the specificity of FDRxLYL as CLS.

We also explored how W383 substitution to residues other than alanine affects ciliary targeting and enzyme activity (*Figure 2—figure supplement 4*). To do this, we mutated W383 to another aromatic residue (W383F), to several aliphatic residues (W383I, W383L, W383M, W383V), or to acid (W383E) or basic (W383R) residues. Interestingly, W383F was fully active and ciliary, indicating that an aromatic ring at this position suffices for both functions (*Figure 2—figure supplement 4*). In contrast, all other mutations fully suppressed ciliary targeting, like W383A (*Figure 2—figure supplement 4a*). Interestingly, enzyme activity of W383I, W383L, W383M, and W383V was the same as for W383A, suggesting that an aromatic ring in this position is important for both targeting and activity (*Figure 2—figure supplement 4b-c*). Overall, the data in this section show that W383 and FDRxLYL, despite their moderate effects on enzyme activity, function as specific CLSs to target INPP5E to cilia.

## The LLxPIR motif cooperates with the CaaX box to target INPP5E to cilia

Besides W383 and FDRxLYL, the C-terminal CaaX box (641-CSVS-644) also modulates INPP5E ciliary targeting. Although CaaX box deletion causes mistargeting of some INPP5E molecules to the ciliary base or other cellular membranes, CaaX box mutants still accumulate in cilia (*Figure 1c*; *Jacoby et al., 2009*; *Thomas et al., 2014*; *Kösling et al., 2018*). This is somewhat puzzling, as INPP5E ciliary targeting strongly depends on the farnesyl receptor PDE6D (*Thomas et al., 2014*).

In the course of our experiments, we made a serendipitous observation that led us to novel insights into how non-farnesylated INPP5E manages to still accumulate in cilia. Initially, we found that INPP5E's N-terminus (aa 1–273), while being dispensable for ciliary targeting of INPP5E (*Figure 1b–c*), is required for ciliary targeting of INPP5E's MORM mutant (Δ627–644) (*Figure 3e*). We subsequently saw that this effect of the N-terminus is mediated by residues 251–273, and that these residues are strongly required for ciliary targeting of the C641S mutant, in which the farnesylated CaaX box cysteine is replaced by non-farnesylatable serine (*Figure 3a–c*). Thus, while both Δ251–273 and C641S single mutants localized inside cilia, the double (Δ251–273)+C641 S mutant completely failed to do so, despite accumulating at the ciliary base (*Figure 3a–c*). Interestingly, upon careful observation and quantitation, both single mutants already showed a partial loss of intraciliary targeting, aside from their ciliary base accumulation (*Figure 3a–c*). Thus, the single Δ251–273 mutant behaves as previously reported for CaaX mutants (and as shown here for C641S), except that Δ251–273 reduces ciliary targeting even more than C641S (*Figure 3c*). These data indicate that the CaaX box and residues 251–273 cooperate to target INPP5E inside cilia: not only are both sequences required for optimal ciliary targeting of the wild type protein, but their functions are partially redundant, each becoming essential when the other one is missing (*Figure 3a–c*).

Even though both these mutations are outside the phosphatase domain, it remains possible that their effects on ciliary targeting are due to disruption of catalytic domain integrity. To test this, we measured the PI(4,5)P$_2$ 5-phosphatase activity of the aforementioned 274–626 mutant, lacking both the N-terminus (Δ1–273) and the MORM region (Δ627–644). This mutant was fully active, with activity and protein levels indistinguishable from WT (*Figure 3—figure supplement 1*). In contrast, the 288–626 mutant, additionally missing residues 275–287 at the beginning of the catalytic domain, had much less activity and reduced levels (*Figure 3—figure supplement 1*). Therefore, residues 251–273

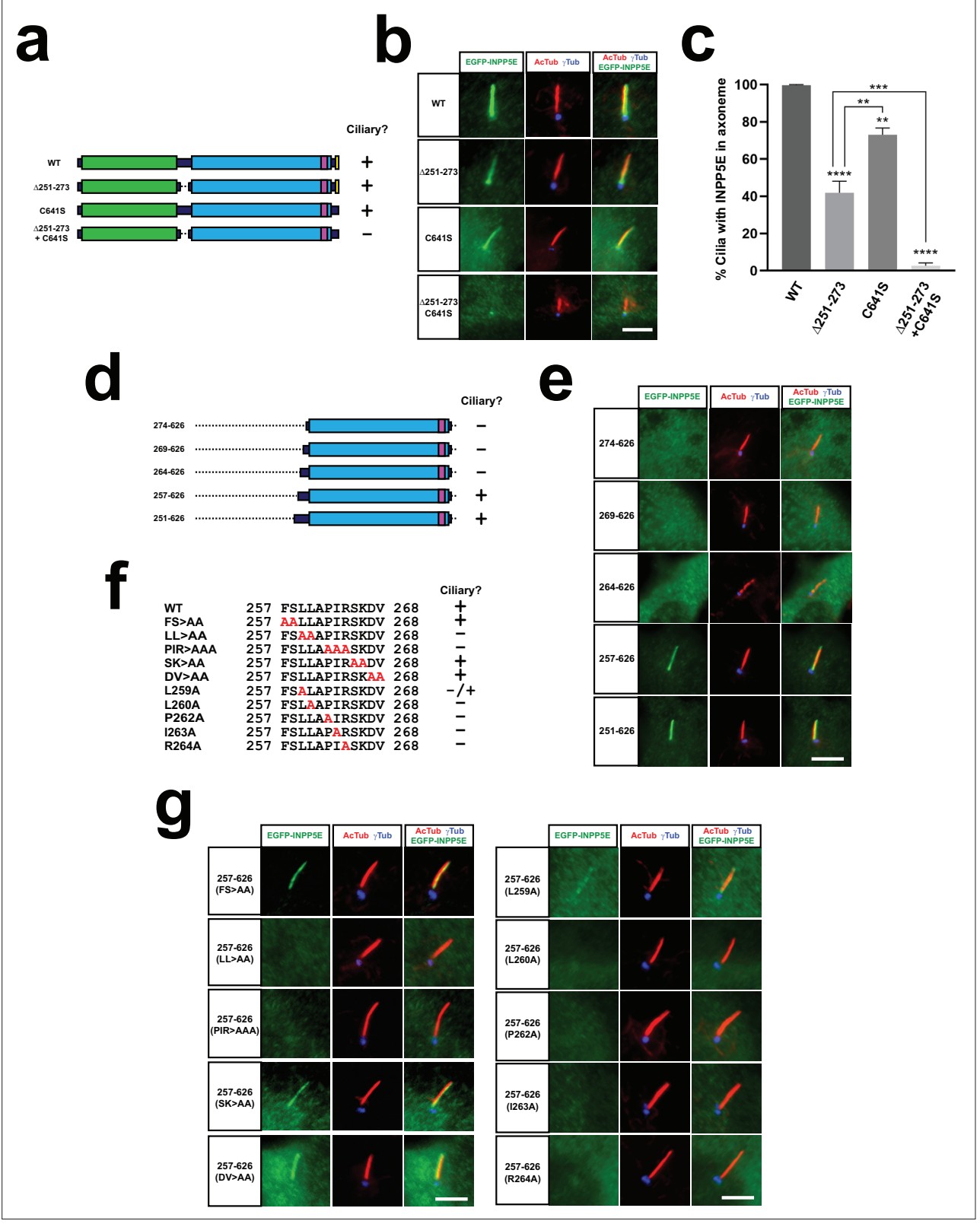

**Figure 3.** The LLxPIR motif is a novel CLS that cooperates with the CaaX box to mediate INPP5E ciliary targeting. (**a**) Schema of full length human INPP5E and its mutants used in (**b–c**). Cilia localization of each mutant is indicated on the right. (**b**) Cilia localization of WT and indicated mutants was analyzed in hTERT-RPE1 cells as in *Figures 1–2*. Images are representative of n=3 independent experiments, with >30 transfected-cell cilia visualized per construct and experiment. Scale bar, 5 µm. (**c**) Quantitation of data from (**b**). The percentage of positive cilia in transfected cells is shown for the

*Figure 3 continued on next page*

*Figure 3 continued*

indicated EGFP-INPP5E constructs. Data are mean ± SEM of n=3 independent experiments. Data were analyzed by one-way ANOVA with post-hoc Tukey multiple comparisons tests. Statistical significance is depicted as p<0.01(**), p<0.001(***), or p<0.0001(****). Significance is shown relative to WT unless otherwise indicated. (**d**) Schema of INPP5E deletion mutants used to map the CLS within aa 251–273. None of these mutants contains the CaaX box (aa 641–644), so their ciliary targeting is strictly dependent on residues 251–273. Cilia localization of each mutant is indicated on the right. (**e**) Cilia localization of the mutants from (**d**) was analyzed in hTERT-RPE1 cells as in *Figures 1–3*. (**f**) Sequence of aa 257–268 in wild type INPP5E and indicated mutants, whose ciliary localization in shown on the right. (**g**) Ciliary targeting of INPP5E(257-626) containing the mutations from (**f**) was analyzed as in (**e**). In both cases, images are representative of n=2 independent experiments, with >30 transfected-cell cilia visualized per construct and experiment. Scale bars, 5 μm.

The online version of this article includes the following source data and figure supplement(s) for figure 3:

**Source data 1.** Source data for *Figure 3c*.

**Figure supplement 1.** CaaX box and the CLS-containing residues 251–273 do not affect enzyme activity.

**Figure supplement 1—source data 1.** Uncropped immunoblot from *Figure 3—figure supplement 1b*.

**Figure supplement 1—source data 2.** Uncropped immunoblot from *Figure 3—figure supplement 1b* (WB: EGFP).

**Figure supplement 2.** Expression levels of EGFP-INPP5E constructs from *Figure 3*.

**Figure supplement 2—source data 1.** Uncropped immunoblots from *Figure 3—figure supplement 2*.

**Figure supplement 2—source data 2.** Uncropped immunoblot from *Figure 3—figure supplement 2* (WB: EGFP).

**Figure supplement 2—source data 3.** Uncropped immunoblot from *Figure 3—figure supplement 2* (WB: Tubulin).

**Figure supplement 2—source data 4.** Uncropped immunoblot from *Figure 3—figure supplement 2* (WB: EGFP).

**Figure supplement 2—source data 5.** Uncropped immunoblot from *Figure 3—figure supplement 2* (WB: Tubulin).

and 627–644 are completely dispensable for enzyme activity, indicating that their effects on ciliary targeting are not caused by disruptions in the phosphatase domain. This was confirmed by Western blot of HEK293T lysates transfected with (Δ251–273)+C641 S or the single Δ251–273 and C641S mutants: in all cases protein levels were indistinguishable from WT (*Figure 3—figure supplement 2*). Thus, residues 251–273 and the CaaX box also behave as bona fide CLSs.

We then mapped which residues within 251–273 are responsible for CLS function. To do this, we started with the 274–626 mutant, which fails to accumulate in cilia as mentioned above. To this mutant, we gradually added residues to the N-terminus, thus creating four more mutants: 269–626, 264–626, 257–626 and 251–626 (*Figure 3d*). Of these, 269–626 and 264–626 failed to accumulate in cilia, whereas 257–626 and 251–626 readily did so (*Figure 3e*). Next, starting from 257 to 626, we generated five alanine-substitution mutants within the 257-FSLLAPIRSKDV-268 region, removing residues FS, LL, PIR, SK, and DV, respectively (*Figure 3f*). Of these, residues LL and PIR were essential for ciliary targeting, whereas FS, SK and DV were fully dispensable (*Figure 3f–g*). We then created the single residue mutants spanning the 259-LLAPIR-264 motif. All these mutants (L259A, L260A, P262A, I263A, and R264A) abolished cilia localization of 257–626, although weak residual targeting was still observed only for L259A (*Figure 3f–g*). Protein levels of 257–626 and the individual LLxPIR motif point mutants were all similar to WT (*Figure 3—figure supplement 2*). Therefore, the LLxPIR motif is a novel CLS that cooperates with the CaaX box to mediate optimal ciliary targeting of INPP5E.

## INPP5E ciliary targeting is mediated by four conserved CLSs

The above data indicate that INPP5E ciliary targeting depends on four CLSs, which we will heretofore refer to as CLS1 (the LLxPIR motif, aa 259–264), CLS2 (W383), CLS3 (the FDRxLYL motif, aa 609–615), and CLS4 (the CaaX box, aa 641–644). Our data actually show that CLS3 goes beyond the FDRxLYL motif, as deletion of residues 616–621 also interferes with ciliary targeting (*Figure 1b–c*). This is consistent with the original report on CLS3, where residues 619–621 were also shown to modulate ciliary targeting (*Humbert et al., 2012*). Thus, CLS3 spans residues 609–621 in human INPP5E.

Since INPP5E ciliary targeting has been reported in different vertebrate species, we examined whether CLS1-4 are conserved in vertebrate evolution. To do this, we aligned the human INPP5E sequence with those of another mammal (mouse), a bird (crow), a reptile (python), an amphibian (toad), and a fish (zebrafish; *Figure 4a*). From this analysis, it is clear that all four CLSs are highly conserved in vertebrates. For CLS1, the consensus sequence is [VL]LxPIR, with only the first leucine admitting a conservative change (*Figure 4a*; *Figure 3g*). CLS2's tryptophan is fully conserved, and so

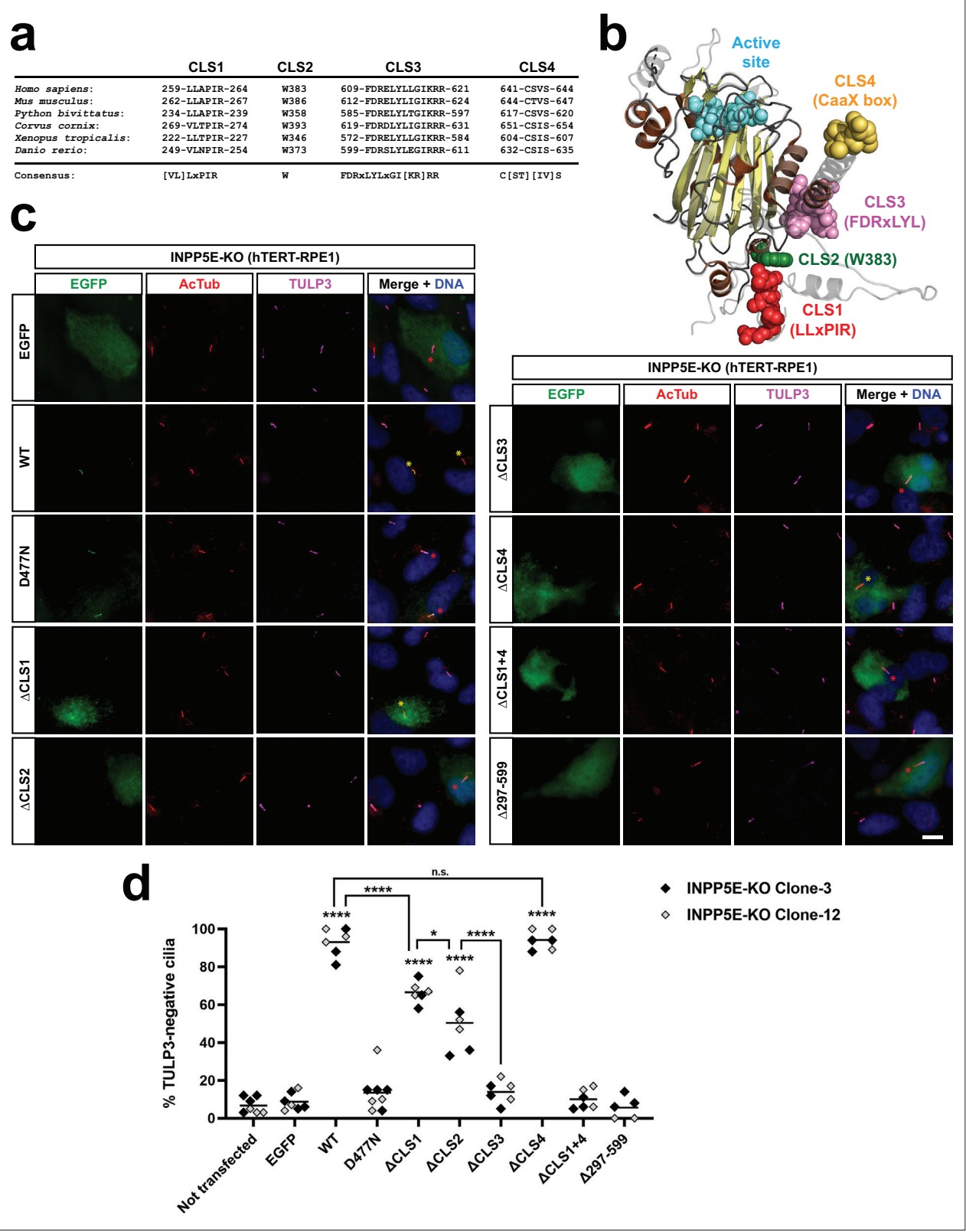

**Figure 4.** CLS1-4 are conserved ciliary localization signals affecting INPP5E function. (**a**) CLS1-4 are highly evolutionarily conserved in vertebrates, including human (NP_063945.2), mouse (AAH80295.1), python (XP_007441606.1), crow (XP_039417670.1), toad (XP_002935265.1), and zebrafish (NP_001096089.2). Consensus sequences are shown below. (**b**) AlphaFold model of INPP5E 3D structure (AF-Q9NRR6-F1) depicting predicted locations of CLS1 (red), CLS2 (green), CLS3 (pink) and CLS4 (yellow). Active site in cyan. Beta-strands and alpha-helices in yellow and brown, respectively. Proline-

*Figure 4 continued on next page*

*Figure 4 continued*

rich N-terminal region (aa 1–200), predicted to be highly flexible, is not shown. CLS1 is probably also part of a flexible region, and its position in the AlphaFold model has a low confidence score (pLDDT). See Uniprot entry Q9NRR6 for more details. (**c**) Rescue assay assessing the ability of INPP5E or its mutants to lower the abnormally high TULP3 levels characteristic of INPP5E-KO cilia. The indicated constructs were transfected into INPP5E-KO RPE1 cells, generated via CRISPR-Cas9 (*Figure 4—figure supplement 2*). Cells were fixed and stained for EGFP, acetylated tubulin (AcTub), TULP3, and DNA (DAPI), as indicated. Scale bar, 10 µm. Note how untransfected INPP5E-KO cells have high ciliary TULP3 levels, as previously described. Transfected cell cilia are labeled with asterisks in the merge panels: yellow asterisks for rescued TULP3-negative cilia, and red asterisks for non-rescued TULP3-positive cilia. (**d**) Quantitation of the rescue experiment shown in (**c**). For each construct, the percentage of TULP3-negative transfected-cell cilia was counted. Data come from five independent experiments. Each point in the graph indicates an independent transfection. Between 12 and 39 transfected-cell cilia were counted per transfection (with exception of the highest data point in ΔCLS2, where only 9 cilia could be counted). Experiments were performed in parallel with two different INPP5E-KO clones (clones 3 and 12). Graph shows individual data points, color-coded by clone as indicated, and the overall median is indicated with a line. Two-way ANOVA revealed significant differences between constructs (p<0.0001) but no significant differences between the clones. All data were then analyzed by one-way ANOVA followed by Tukey tests. Significance is shown relative to EGFP unless otherwise indicated. p<0.0001 (****); p<0.05 (*); not significant (n.s.).

The online version of this article includes the following source data and figure supplement(s) for figure 4:

**Source data 1.** Source data for *Figure 4d*.

**Figure supplement 1.** Generation of puromycin-sensitive hTERT-RPE1 cells by CRISPR/Cas9.

**Figure supplement 2.** Generation of INPP5E-KO hTERT-RPE1 cells by CRISPR/Cas9.

**Figure supplement 3.** CLS1 and CLS4 mutants are only seen at the transition zone in methanol-fixed cells.

is CLS3, as previously shown (consensus: FDRxLYLxGI[KR]RR) (*Figure 4a*; *Humbert et al., 2012*). For CLS4 the consensus is C[ST][IV]S, which in all cases encodes a farnesyl transferase-specific CaaX box (*Wright and Philips, 2006*). Thus, the four CLSs are highly conserved. Moreover, they are all found on the same side of INPP5E according to the AlphaFold model, even though CLS1 location in this model has a low confidence score (*Figure 4b*; *Jumper et al., 2021*).

By affecting INPP5E ciliary targeting, these CLSs are expected to affect INPP5E ciliary function. To test this, we assessed the ability of the CLS mutants to rescue the well-documented abnormal ciliary accumulation of TULP3 in INPP5E-KO cells (*Garcia-Gonzalo et al., 2015*; *Chávez et al., 2015*). We performed this assay using INPP5E-KO hTERT-RPE1 cells, which we generated in two steps. First, by using CRISPR/Cas9 technology, we created puromycin-sensitive hTERT-RPE1 cells to knock out the puromycin acetyltransferase (*PAC*) gene, stably inserted into these cells upon their immortalization (*Figure 4—figure supplement 1a*). As expected, the PAC-KO clones thus obtained were highly sensitive to puromycin, in contrast to the original hTERT-RPE1 cells (*Figure 4—figure supplement 1b-c*). In the second step, using puromycin selection-based CRISPR on these cells, we generated INPP5E-KO cells, as confirmed by genomic analysis (*Figure 4—figure supplement 2a*). Immunofluorescence with anti-INPP5E antibodies confirmed the absence of INPP5E from these cells, despite some non-specific non-ciliary staining (*Figure 4—figure supplement 2b*). As expected, anti-TULP3 antibodies revealed a massive accumulation of this protein in the INPP5E-KO cilia (*Figure 4—figure supplement 2c*).

In non-transfected or EGFP-transfected INPP5E-KO cells, the percentage of TULP3-negative cilia was very low (≈10%) (*Figure 4c–d*). As shown previously in a similar assay using Inpp5e-null mouse embryonic fibroblasts (MEFs; *Garcia-Gonzalo et al., 2015*), transfection of EGFP-INPP5E WT into our INPP5E-KO cells effectively rescued the phenotype, driving the percentage of TULP3-negative cilia to ≈90%. Also as previously reported, the catalytically inactive D477N mutant completely failed to rescue (*Figure 4c–d*; *Garcia-Gonzalo et al., 2015*). Having validated the assay, we assessed rescue by the following CLS mutants: Δ251–273 (ΔCLS1), W383A (ΔCLS2), F609A+D610A+R611 A (ΔCLS3), C641S (ΔCLS4), the double ΔCLS1+ΔCLS4 mutant (ΔCLS1 +4), and the catalytic domain deletion Δ297–599. Interestingly, ΔCLS4 was fully functional in this assay, ΔCLS1 a little less so, and ΔCLS1 +4 had no rescue activity whatsoever (*Figure 4c–d*). Thus, the relationship between CLS1 and CLS4 in this assay largely mirrors the redundancy observed in their ciliary targeting (*Figure 3c*). As expected, ΔCLS3 and Δ297–599 were also completely inactive (*Figure 4c–d*). To our surprise, ΔCLS2 showed partial rescue: less than ΔCLS1, but more than ΔCLS3 (*Figure 4c–d*). The reasons for this partial rescue are still unclear to us and will require further investigation. In any case, it remains true that all non-ciliary CLS mutants showed strongly reduced functionality in the TULP3 rescue assay.

We then examined how CLS1-4 affect INPP5E cilia localization as determined by methanol fixation of hTERT-RPE1 cells (rather than fixing first with paraformaldehyde (PFA), as done so far). We did

this on the basis of a recent study showing that, in these cells, live imaging of GFP-INPP5E shows a strong accumulation at the ciliary base (in addition to some intraciliary staining), and that this ciliary base accumulation is best recapitulated with methanol-only fixation (whereas PFA is better suited to visualize the intraciliary staining) (*Conduit et al., 2021*). Indeed, with methanol fixation, EGFP-INPP5E WT was clearly accumulated at the base of most cilia (≈60%), with the remaining cilia showing a more uniform distribution along the axoneme (≈40%) (*Figure 4—figure supplement 3a-b*). Regardless of cilia base-enrichment, some intraciliary axonemal staining was always seen with EGFP-INPP5E WT under these conditions. In contrast, no axonemal staining was ever observed for any of the CLS mutants (ΔCLS1, ΔCLS2, ΔCLS3, ΔCLS4, or ΔCLS1+4) (*Figure 4—figure supplement 3a-b*). This confirms the importance of all these CLSs for INPP5E to reach inside cilia. It also shows that the axonemal staining seen with ΔCLS1 and ΔCLS4 in PFA-fixed cells (*Figure 3b–c*) is lost with the methanol-only fixation (*Figure 4—figure supplement 3a-b*). Interestingly, with the latter fixation, ΔCLS1, ΔCLS4, and ΔCLS1+4 were all strongly accumulated at the ciliary base (in 100% of cilia). These accumulation occurred at the ciliary transition zone, distal from basal body and proximal from axoneme, as revealed by signal intensity scans along these cilia (*Figure 4—figure supplement 3c*). In contrast, ΔCLS3, was never seen at the ciliary base, whereas ΔCLS2 was occasionally seen there (30% of cilia) (*Figure 4—figure supplement 3a-b*). Altogether, the data in this section confirm the importance of INPP5E's four CLSs.

## Ciliary targeting is affected by some INPP5E ciliopathy mutations

INPP5E ciliary targeting is dependent on CLS1, CLS2, CLS3, and CLS4 (*Figures 2–4*), and on the integrity of its catalytic domain (*Figure 1*). The MORM mutation (Δ627–644), which deletes CLS4 and moderately reduces cilia localization, as mentioned above, is the only known INPP5E ciliopathy mutation affecting its ciliary targeting. However, whether and how other ciliopathy mutations in INPP5E also affect ciliary targeting is largely unknown. To address this, we sought ciliopathy mutations locating near a CLS, or that were likely to compromise catalytic domain integrity. In this manner, we identified twelve mutations, eleven from JBTS and one from LCA-like disease (*Figure 5a–b*; *Bachmann-Gagescu et al., 2015*; *Bielas et al., 2009*; *Travaglini et al., 2013*; *de Goede et al., 2016*; *Xu et al., 2015*; *Wang et al., 2013*).

One of the mutations, C641R, replaces the farnesylatable cysteine in CLS4 by an arginine. Not surprisingly, this mutation affected ciliary targeting in the same way as C641S or MORM, and did not affect protein levels (*Figure 5b–d*). Several other mutations affected residues located near the CLS2-CLS3 region. These included R345S, T355M, R378C, C385Y, V388L, R621Q, and R621W. None of these affected ciliary localization, despite causing moderate reductions in protein levels (*Figure 5b–d and g*). Since almost all INPP5E ciliopathy mutations affect the catalytic domain, none were found near CLS1. The closest one to CLS1 was G286R, at the beginning of the catalytic domain. Interestingly, G286R largely abolished ciliary targeting, although this is more likely due to its affecting catalytic domain integrity (*Figure 5b–e*). We also tested mutations, like W474R and V303M, that we thought likely to disrupt the catalytic domain, given their location at the core of the beta-sandwich. Indeed, W474R abolished ciliary targeting, and V303M reduced it considerably, with levels of both reduced (*Figure 5b–d and f*). Since W474R and G286R were found in homozygosis in JBTS patients, this suggests that some JBTS patients cannot target INPP5E to cilia (*Figure 5a*). Finally, we also tested the D490Y mutation, located near a beta-hairpin close to the active site, but far from any CLS. D490Y did not affect ciliary targeting, despite a strong reduction in protein levels (*Figure 5b–e*). Altogether, these data show that INPP5E ciliary targeting is sometimes affected in Joubert syndrome, which could contribute to pathogenesis in these cases.

## INPP5E binding to PDE6D is CLS4-dependent

After identifying novel INPP5E CLSs and exploring their role in ciliopathies, we focused on the mechanisms of action of these CLSs. Presumably, these CLSs act by binding to other proteins implicated in INPP5E ciliary targeting. One such protein is PDE6D, a ciliary cargo receptor for prenylated proteins. Although PDE6D binding to INPP5E is CLS4-dependent and CLS3-independent, whether its interaction requires our newly identified CLSs (CLS1 and CLS2) is unknown (*Thomas et al., 2014*; *Humbert et al., 2012*; *Qiu et al., 2021*; *Rao et al., 2016*). To test this, we cotransfected HEK293T cells with plasmids encoding Flag-PDE6D and EGFP-INPP5E in order to perform coimmunoprecipitation (co-IP)

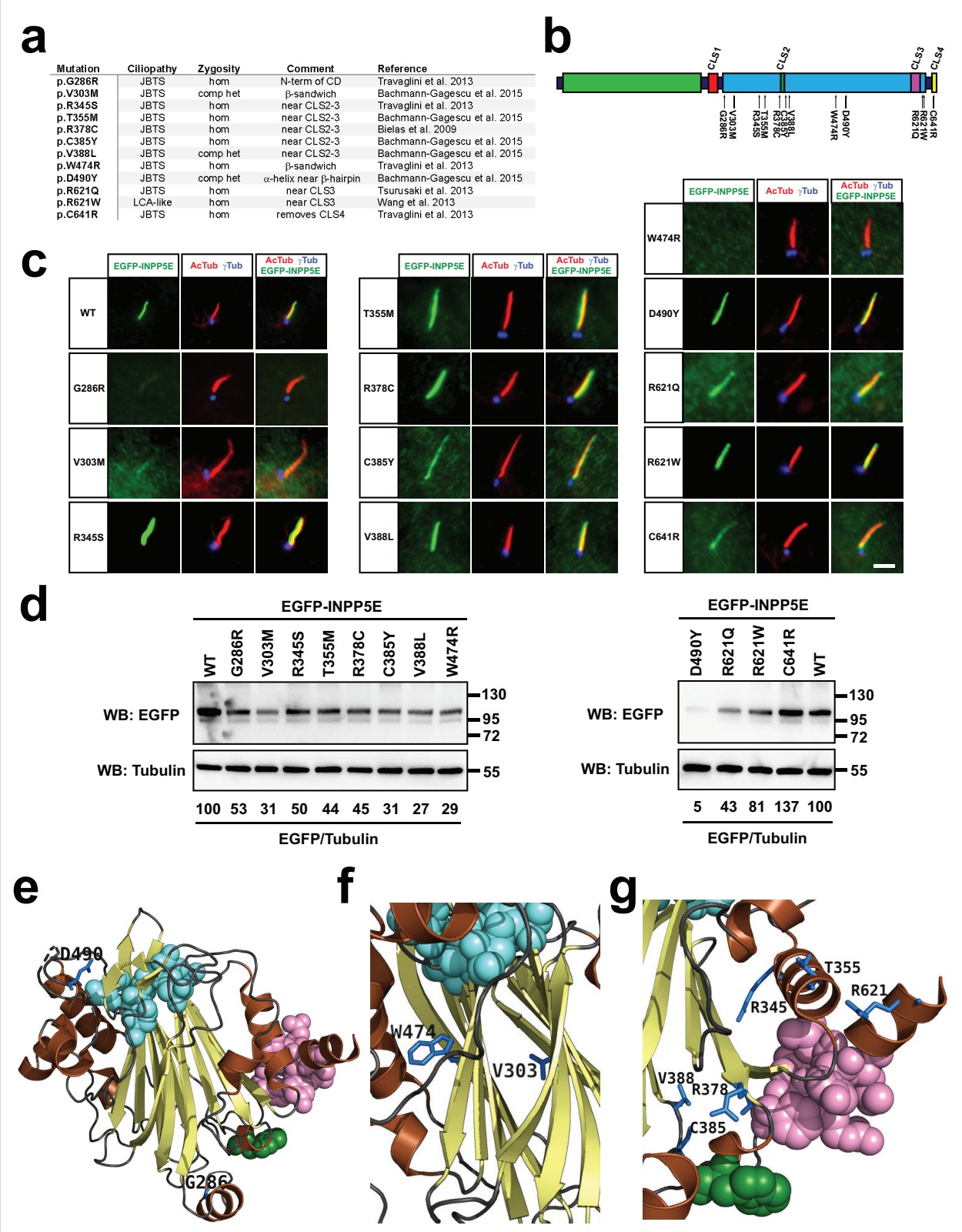

**Figure 5.** A subset of Joubert syndrome INPP5E mutations abolishes ciliary targeting. (**a**) Table of INPP5E ciliopathy mutations analyzed here. JBTS: Joubert syndrome; LCA: Leber congenital amaurosis; hom: homozygous; comp het: compound heterozygous. (**b**) Schema of INPP5E protein sequence indicating the locations of the ciliopathy mutations from (**c**) relative to its four CLSs, its catalytic domain (cyan) and its N-terminal proline-rich region (green). (**c**) Ciliary localization of mutants from (**a–b**) was analyzed in hTERT-RPE1 cells as in *Figures 1–3*. Images are representative of at least two

*Figure 5 continued*

independent experiments per construct, with >30 transfected-cell cilia visualized per construct and experiment. Scale bar, 5 µm. (**d**) The mutants from (**a–c**) were expressed in HEK293T cells and their protein levels analyzed by SDS-PAGE and immunoblotting with anti-EGFP antibody, and anti-alpha tubulin as loading control. Molecular weight markers in kilodaltons are shown on the right. The numbers under the tubulin blots are EGFP/Tubulin band intensity ratios, normalized so that WT equals 100%. (**e–g**) 3D views of INPP5E catalytic domain (PDB ID: 2xsw) showing the ciliopathy-mutated residues from (**a**) in dark blue (other colors as in *Figure 4b*). (**e**) Full catalytic domain showing G286 (bottom) and D490 (top left). (**f**) closeup view of beta-sandwich showing W474 and V303. (**g**) closeup view of CLS2-3 region showing R345, T355, R378, C385, V388L, and R621.

The online version of this article includes the following source data for figure 5:

**Source data 1.** Uncropped immunoblots from *Figure 5d*.

**Source data 2.** Uncropped immunoblot from *Figure 5d* (WB: EGFP).

**Source data 3.** Uncropped immunoblot from *Figure 5d* (WB: Tubulin).

**Source data 4.** Uncropped immunoblot from *Figure 5d* (WB: EGFP).

**Source data 5.** Uncropped immunoblot from *Figure 5d* (WB: Tubulin).

experiments (*Figure 6a*). As expected, Flag-PDE6D robustly co-immunoprecipitated (co-IPed) with EGFP-INPP5E(WT), but not EGFP control (*Figure 6a*). This interaction was completely dependent on CLS4, and completely independent of CLS1, CLS2, and CLS3 (*Figure 6a*). Accordingly, ΔCLS1 +4 behaved the same as ΔCLS4, with virtually no co-IP observed, whereas ΔCLS2 and ΔCLS3 co-IPed the same as WT (*Figure 6a*). We also tested whether the co-IP involved INPP5E's N-terminal (aa 1–283) or C-terminal (251-644) regions, with the latter being the case for PDE6D (*Figure 6a*).

In addition to co-IP, we also studied the INPP5E-PDE6D interaction in vivo by means of co-recruitment assays (*Figure 6e–f*). To do this, we generated bait constructs expressing mCherry-FKBP-INPP5E fusion proteins (WT, ΔCLS1, ΔCLS2, ΔCLS3, ΔCLS4, or ΔCLS1+4), or mCherry-FKBP as negative control. Each of these bait constructs was separately co-expressed in HeLa cells with the prey construct (mVenus-PDE6D), and with FRB-CFP-CaaX, whose prenylated CaaX box tethers FRB to the inner leaflet of the plasma membrane (*Figure 6e*). Upon inducing the FKBP-FRB interaction with rapamycin, this should recruit bait constructs to the plasma membrane, a recruitment that can be monitored and quantitated by total internal reflection fluorescence (TIRF) microscopy (*Figure 6e–f*; *Lin et al., 2013*; *Gallego et al., 2013*). Additionally, if bait and prey interact, then prey co-recruitment to the plasma membrane will also be observed (*Figure 6e–f*). As expected, rapamycin-induced robust plasma membrane recruitment of all bait constructs (*Figure 6g*). Also as expected, mVenus-PDE6D co-recruitment was much higher with mCherry-FKBP-INPP5E(WT) than with the mCherry-FKBP control, confirming the specificity of PDE6D-INPP5E binding (*Figure 6g–h*). As observed in the co-IPs, mVenus-PDE6D co-recruitment was strongly reduced by both the ΔCLS4 and ΔCLS1+4 mutations, indicating a strong CLS4-dependence (*Figure 6g–h*). Also consistent with the co-IPs, ΔCLS1 and ΔCLS3 did not affect mVenus-PDE6D co-recruitment, whereas ΔCLS2 caused a modest reduction (*Figure 6g–h*). Altogether, the co-IP and in vivo co-recruitment data demonstrate that CLS4 is the key CLS controlling INPP5E-PDE6D binding.

## INPP5E binding to RPGR is CLS4-dependent

RPGR also interacts with INPP5E and is required for its ciliary targeting (*Rao et al., 2016*). Moreover, RPGR ciliary targeting is also dependent on PDE6D, which binds to both its geranylgeranylated CaaX box and its RCC1-like domain (*Zhang et al., 2019*; *Rao et al., 2016*; *Dutta and Seo, 2016*; *Lee and Seo, 2015*; *Fansa et al., 2015*; *Wätzlich et al., 2013*; *Vössing et al., 2021*). Despite all these connections, how RPGR mediates INPP5E ciliary targeting is unclear. To address this, we performed co-IPs between Flag-RPGR and the same EGFP-INPP5E constructs used above for the PDE6D co-IPs. As with PDE6D, the INPP5E-RPGR co-IP was abolished by ΔCLS4 and ΔCLS1+4, but was untouched by ΔCLS1, ΔCLS2 or ΔCLS3 (*Figure 6b*). Therefore, INPP5E's farnesylated CaaX box is also key for its association to RPGR. Consistently, RPGR strongly co-IPed with INPP5E's C-terminal fragment (251-644) but not with the N-terminal one (1-283).

## INPP5E binding to ARL13B is promoted by CLS2, CLS3, and CLS4

ARL13B is another key mediator of INPP5E ciliary targeting (*Humbert et al., 2012*; *Fujisawa et al., 2021*; *Qiu et al., 2021*). ARL13B regulates INPP5E in at least two different ways. First, ARL13B acts

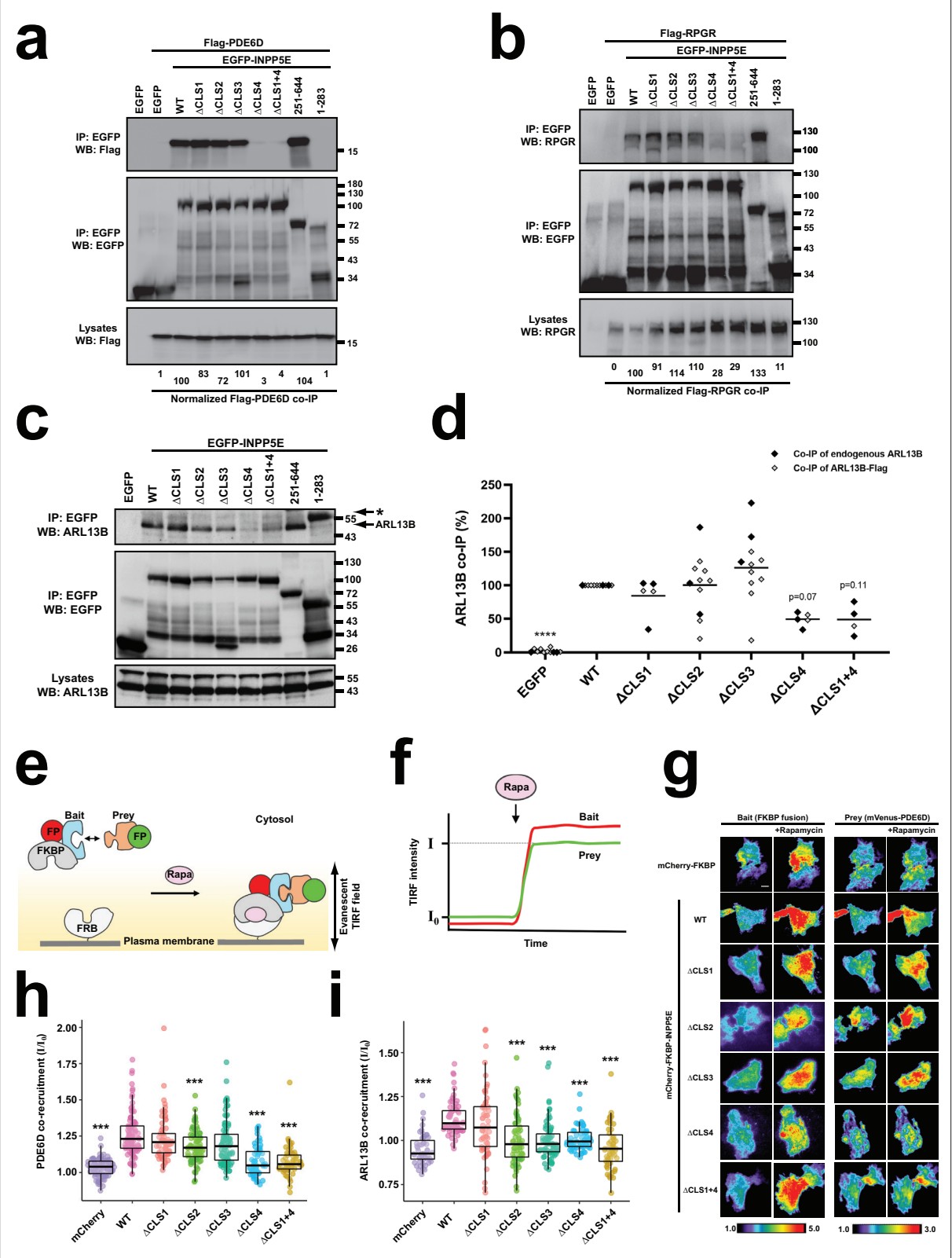

**Figure 6.** CLS4 promotes INPP5E binding to PDE6D, RPGR and ARL13B. (**a**) The indicated EGFP-INPP5E variants were coexpressed in HEK293T cells with Flag-PDE6D, as indicated. Lysates were immunoprecipitated with GFP-Trap beads and analyzed by Western blot with the indicated antibodies. Molecular weight markers in kilodaltons are shown on the right. Quantitation of Flag-PDE6D co-immunoprecipitation (co-IP), as percentage relative to WT, is shown at the bottom. Quantitations are normalized relative to both immunoprecipitated EGFP constructs and lysate amounts of Flag-PDE6D. (**b**)

*Figure 6 continued on next page*

*Figure 6 continued*

Same experiment as in (**a**) but Flag-RPGR was used instead of Flag-PDE6D. Quantitations at the bottom are normalized relative to immunoprecipitated EGFP constructs. (**c**) Coimmunoprecipitation of endogenous ARL13B with the indicated EGFP-INPP5E constructs in HEK293T cells. Asterisk points to EGFP-INPP5E(1-283) band. (**d**) Quantitation of ARL13B co-IP with the indicated EGFP-INPP5E constructs from n=11 independent experiments in HEK293T cells. Black and grey dots correspond, respectively, to experiments where endogenous ARL13B or exogenous ARL13B-Flag co-IP was assessed. Not all samples were present in all experiments (but EGFP, WT, ΔCLS2 and ΔCLS3 were). Two-way ANOVA revealed significant differences between constructs (p<0.0001) but no significant differences between using endogenous or exogenous ARL13B. All data were then analyzed by one-way ANOVA followed by Tukey tests. Significance is shown relative to WT: p<0.0001 (****). (**e**) Schema of chemically-inducible co-recruitment assay. Rapamycin (Rapa)-induced interaction between FKBP and FRB is used to quantitate binding of prey candidates to a bait. FKBP is fused to the prey along with a fluorescent protein, while FRB is tethered to inner leaflet of plasma membrane. Upon rapamycin addition, FKBP binds to FRB, bringing bait (red FP) and associated prey (green FP) to the plasma membrane. (**f**) Recruitment of bait and prey to plasma membrane can be sensitively detected by TIRF microscopy as an increased fluorescence signal. The ratio of final to initial TIRF intensity upon rapamycin addition ($I/I_0$) for the prey provides a quantitative measure of prey's co-recruitment to plasma membrane by bait, and hence of the prey-bait interaction. (**g**) TIRF microscopy images showing rapamycin-induced plasma membrane recruitment of bait constructs (left) and the corresponding co-recruitment of prey (mVenus-PDE6D, right). Intensity scales are depicted at bottom. Scale bar, 10 µm. (**h**) Normalized rapamycin-induced co-recruitment of mVenus-PDE6D (prey) by mCherry-FKBP-INPP5E (WT or indicated mutants), or by mCherry-FKBP (mCherry) as negative control. Individual measurements of n>50 cells per condition are shown. Box and whisker plots represent median, first and third quartiles, and 95% confidence intervals. Statistical significance relative to WT is shown as *** p<0.001 (unpaired Student's t-tests). (**i**) Normalized rapamycin-induced co-recruitment of ARL13B-EYFP (prey) by mCherry-FKBP-INPP5E (WT or indicated mutants), or by mCherry-FKBP (mCherry) as negative control. Data acquisition, analysis and representation as in (**h**).

The online version of this article includes the following source data for figure 6:

**Source data 1.** Uncropped immunoblots from *Figure 6*.

**Source data 2.** Uncropped immunoblot from *Figure 6a* (IP: EGFP; WB: Flag).

**Source data 3.** Uncropped immunoblot from *Figure 6a* (IP: EGFP; WB: EGFP).

**Source data 4.** Uncropped immunoblot from *Figure 6a* (Lysates, WB: Flag).

**Source data 5.** Uncropped immunoblot from *Figure 6b* (IP: EGFP; WB: RPGR).

**Source data 6.** Uncropped immunoblot from *Figure 6b* (IP: EGFP; WB: EGFP).

**Source data 7.** Uncropped immunoblot from *Figure 6b* (Lysates, WB: RPGR).

**Source data 8.** Uncropped immunoblot from *Figure 6c* (IP: EGFP; WB: ARL13B).

**Source data 9.** Uncropped immunoblot from *Figure 6c* (IP: EGFP; WB: EGFP).

**Source data 10.** Uncropped immunoblot from *Figure 6c* (Lysates, WB: ARL13B).

**Source data 11.** Source data from *Figure 6d*.

as a guanine nucleotide exchange factor (GEF) for ARL3, whose active GTP-bound form promotes dissociation of the PDE6D-INPP5E complex after it reaches the ciliary lumen (*Gotthardt et al., 2015*; *Fansa et al., 2016*; *Ivanova et al., 2017*; *ElMaghloob et al., 2021*). Additionally, ARL13B directly interacts with INPP5E and is required for its ciliary retention (*Humbert et al., 2012*; *Fujisawa et al., 2021*; *Qiu et al., 2021*). Since the first mechanism is mediated by ARL3, we checked whether INPP5E binds ARL3. However, we detected no co-IP between EGFP-INPP5E and ARL3-myc, or its constitutively active form ARL3(Q71L)-myc, in accordance with previous data (data not shown; *Humbert et al., 2012*). Likewise, we found no interaction between INPP5E and BART, a protein that cooperates with ARL13B as a co-GEF for ARL3 (data not shown; *ElMaghloob et al., 2021*).

We then carried out co-IPs to assess the CLS-dependence of the ARL13B-INPP5E interaction. We readily detected co-IP of endogenous ARL13B with EGFP-INPP5E in HEK293T cells. This co-IP was much lower with ΔCLS4 and ΔCLS1+4, but not clearly affected by CLS1-3 (*Figure 6c*). As with PDE6D and RPGR, INPP5E's C-terminal region (251-644), but not the N-terminal (1-283), sufficed for the INPP5E-ARL13B interaction (*Figure 6c*). Since CLS3 was previously reported to affect INPP5E-ARL13B co-IP, we examined this question further (*Humbert et al., 2012*). In total, we performed 11 co-IP experiments between these proteins. In three of them (including the one in *Figure 6c*), we measured co-IP of endogenous ARL13B. In the other eight, we measured co-IP of co-transfected ARL13B-Flag. Quantitation of all these experiments showed no discernible effect of CLS1-3 on ARL13B co-IP, while largely confirming the effect of CLS4 (*Figure 6d*). Since the previously reported effect had been observed using deletions (ΔFDR, ΔLYL) rather than alanine-substitutions (FDR >AAA, LYL >AAA) of the CLS3 residues, we also performed an experiment comparing ARL13B-Flag co-IP for all these four mutants. This experiment showed no differences among any of these mutants, all of which co-IPed

ARL13B like WT control (data not shown). Thus, we conclude that CLS4, but not CLS1-3, affect ARL13B co-IP by INPP5E in our experiments.

We also studied INPP5E-ARL13B association by means of co-recruitment assays like those in *Figure 6e–h*. In this case, ARL13B-EYFP was used as prey instead of mVenus-PDE6D (*Figure 6i*). ARL13B-EYFP co-recruitment strongly increased with mCherry-FKBP-INPP5E(WT), as compared to mCherry-FKBP alone, demonstrating a specific interaction. This interaction was unaffected by ΔCLS1 but was strongly reduced with ΔCLS2, ΔCLS3, ΔCLS4 and ΔCLS1+4 (*Figure 6i*). Therefore, in addition to confirming the key role of CLS4, the in vivo co-recruitment data showed that CLS2-3 do affect INPP5E-ARL13B association after all. The reason why the effects of CLS2-3 are readily detectable in our co-recruitment but not out co-IP assays remains to be determined. Taken together, our data indicate that CLS2, CLS3 and CLS4 all promote INPP5E association with ARL13B.

## INPP5E binding to TULP3 is promoted by CLS2 and CLS3

TULP3 is a ciliary trafficking adapter needed for ciliary targeting of membrane proteins such as G-protein-coupled receptors, polycystins, ARL13B and INPP5E (*Badgandi et al., 2017*; *Han et al., 2019*; *Mukhopadhyay et al., 2010*; *Mukhopadhyay et al., 2013*; *Legué and Liem, 2019*; *Hilgendorf et al., 2019*; *Hwang et al., 2019*). However, whether INPP5E and TULP3 interact is not known. We therefore tested this. Indeed, EGFP-INPP5E specifically co-IPed TULP3-myc (*Figure 7a*). Such co-IP was unaffected by ΔCLS1, ΔCLS4, and ΔCLS1+4, but was clearly reduced by ΔCLS2 and ΔCLS3 (*Figure 7a–b*). Consistently, TULP3 interacted strongly with EGFP-INPP5E(251-644), and much less so with EGFP-INPP5E(1-283) (*Figure 7a*).

TULP3 functions as an adapter by connecting the IFT trafficking machinery (which it binds via its N-terminal domain, NTD: aa 1–183) to membrane proteins (which it binds via its phosphoinositide-binding C-terminal Tubby domain, CTD: aa 184–442) (*Mukhopadhyay et al., 2010*). To test how TULP3 binds INPP5E, we assessed co-IP of Flag-INPP5E by different EGFP-TULP3 constructs (*Figure 7c*). Flag-INPP5E specifically co-IPed with full length TULP3, an interaction that was largely dependent on TULP3's CTD, even though weak binding to NTD was also observed (*Figure 7c*). In addition, we tested Flag-INPP5E binding to two TULP3 mutants, namely K268A+R270 A and K389A. The former cannot bind phosphoinositides, and may also be hypoacetylated (*Mukhopadhyay et al., 2010*; *Kerek et al., 2021*), whereas the latter removes a key lysine needed for TULP3 to interact with ARL13B and target it to cilia, according to a recent preprint (*Palicharla et al., 2021*). Interestingly, both mutations clearly reduced INPP5E binding, though none abolished it completely (*Figure 7c*). Therefore, our data show that INPP5E interacts with TULP3, and that this interaction is dependent on: (i) CLS2 and CLS3 in INPP5E's catalytic domain, and (ii) TULP3's Tubby domain and its ability to bind ARL13B and phosphoinositides (*Figure 7d*).

## INPP5E-CEP164 interaction is downregulated by CLS2-3

INPP5E also interacts with CEP164, a ciliary base protein essential for ciliogenesis. In CEP164-silenced non-ciliated cells, INPP5E fails to accumulate at the centrosome (*Humbert et al., 2012*). This suggests that CEP164, by recruiting INPP5E to the ciliary base, may contribute to its ciliary targeting. Because of this, we also examined the INPP5E-CEP164 interaction. To do this, we looked at how EGFP-INPP5E and its mutants co-IP endogenous CEP164 in HEK293T cells. Interestingly, while WT, ΔCLS1, ΔCLS4 and ΔCLS1+4 all co-IPed similar amounts of CEP164, the ΔCLS2 and ΔCLS3 mutants displayed a stronger interaction, suggesting that CLS2 and CLS3 downregulate CEP164 binding (*Figure 7e*). This effect of CLS2-3 was confirmed after quantitation of five similar experiments (*Figure 7f*). Moreover, CEP164 strongly interacted with INPP5E's N-terminal fragment (1-283), but not with the CLS2/3-containing C-terminal one (251-644; *Figure 7e*). Since much more CEP164 was pulled down by EGFP-INPP5E(1-283) than EGFP-INPP5E(WT), despite both fusion proteins being expressed similarly, this further indicates that CLS2-3 antagonize CEP164 binding, which is mediated by INPP5E's N-terminal region (*Figure 7e*).

CEP164 contains a WW domain near its N-terminus and several coiled coils in the rest of its long sequence (1460 aa) (*Cajanek and Nigg, 2014*; *Schmidt et al., 2012*; *Graser et al., 2007*). Since WW domains, like SH3 domains, interact with proline-rich ligands, we hypothesized that CEP164's WW might interact with INPP5E's proline-rich N-terminus (*Macias et al., 2002*). To test this, we first checked whether Flag-INPP5E was co-IPed by three CEP164 fragments spanning its N-terminal (NT,

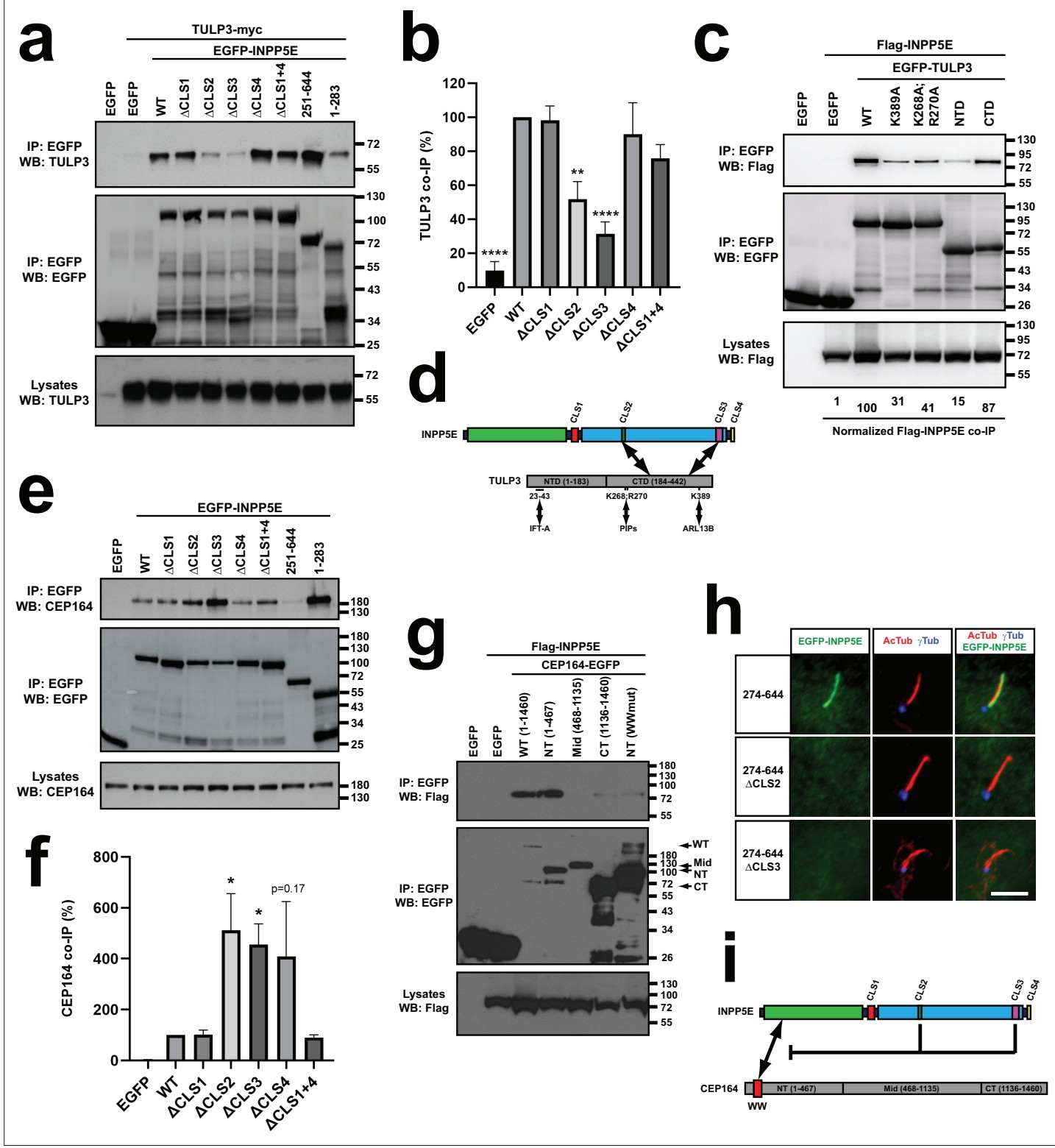

**Figure 7.** CLS2 and CLS3 regulate INPP5E binding to TULP3 and CEP164. (**a**) The indicated EGFP-INPP5E variants were coexpressed in HEK293T cells with TULP3-myc as indicated. Lysates were immunoprecipitated with GFP-Trap beads and analyzed by Western blot with TULP3 and EGFP antibodies, as indicated. (**b**) Quantitation of TULP3-myc co-IP by the indicated EGFP-INPP5E constructs in HEK293T cells. Co-IP levels, expressed as percentage of WT, are normalized by the amounts of both immunoprecipitated EGFP-INPP5Es and TULP3-myc lysate levels. Data are mean ± SEM of n=7,7,4,7,7,4,4 independent experiments and were analyzed by one-way ANOVA followed by Tukey multiple comparisons tests. Significance relative to WT is shown

*Figure 7 continued on next page*

*Figure 7 continued*

as p<0.01(**) and p<0.0001(****). (**c**) The indicated EGFP-TULP3 variants were coexpressed in HEK293T cells with Flag-INPP5E as indicated. Lysates were immunoprecipitated with GFP-Trap beads and analyzed by Western blot with the indicated antibodies. NTD: N-terminal domain (aa 1–183); CTD: C-terminal Tubby domain (aa 184–442). Numbers at the bottom show quantitation of Flag-INPP5E co-IP as percentage of WT, normalized by both immunoprecipitated EGFP-TULP3, and by Flag-INPP5E lysate levels. (**d**) Schema of INPP5E-TULP3 interaction. On INPP5E's side, the interaction mostly involves the catalytic domain, requiring CLS2 and CLS3. On TULP3's side, the interaction occurs mostly through the CTD and is affected by the ARL13B-binding K389, and by the phosphoinositide (PIPs)-binding K268 and R270. (**e**) Lysates of HEK293T cells expressing the indicated EGFP-INPP5E variants were immunoprecipitated with GFP-Trap beads and the levels of endogenous CEP164 and exogenous EGFP were analyzed by Western blot as indicated. Molecular weight markers on the right. (**f**) Quantitation of endogenous CEP164 co-IP by the indicated EGFP-INPP5E constructs in HEK293T cells. Co-IP levels were calculated and plotted as in (**b**). Data are mean ± SEM of n=5,5,3,5,5,3,3 independent experiments and were analyzed by one-way ANOVA followed by Dunnett multiple comparisons tests relative to WT. Significance is shown as p<0.05(*). (**g**) Flag-INPP5E was coexpressed in HEK293T cells with the indicated CEP164-EGFP variants, including full length CEP164 (aa 1–1460), its N-terminal (NT, 1–467), middle (Mid, 468–1135) and C-terminal (CT, 1136–1460) regions, and NT carrying a mutated WW domain (WW: aa 56–89; mutation: Y73A+Y74 A). Arrows indicate the positions of these proteins. Lysates were immunoprecipitated with GFP-Trap beads and analyzed by Western blot with antibodies against Flag or EGFP, as indicated. Molecular weight markers are displayed on the right. (**h**) CLS2 and CLS3 are still required for INPP5E ciliary targeting in mutants unable to bind CEP164. Cilia localization was analyzed as in previous figures for the indicated EGFP-INPP5E variants, all of which lack aa 1–273 and hence cannot bind CEP164. Images are representative of n=2 independent experiments, with >30 transfected-cell cilia visualized per construct and experiment. Scale bar, 5 μm. (**i**) Schema summarizing results from (**e–h**). CEP164-NT is sufficient for INPP5E binding provided the WW domain is intact. On INPP5E's side, the proline-rich N-terminal region (aa 1–283) is sufficient to interact with CEP164. Moreover, INPP5E(1-283), INPP5E(ΔCLS2) and INPP5E(ΔCLS3) mutants all bind CEP164 more intensely than INPP5E(WT), indicating that INPP5E's C-terminal region downregulates CEP164 binding in a CLS2/3-dependent manner. This may or may not be necessary for INPP5E ciliary targeting, but it is clearly not sufficient, as shown by the data in (**h**).

The online version of this article includes the following source data and figure supplement(s) for figure 7:

**Source data 1.** Uncropped immunoblots from *Figure 7*.

**Source data 2.** Uncropped immunoblot from *Figure 7a* (IP: EGFP; WB: TULP3).

**Source data 3.** Uncropped immunoblot from *Figure 7a* (IP: EGFP; WB: EGFP).

**Source data 4.** Uncropped immunoblot from *Figure 7a* (Lysates, WB: TULP3).

**Source data 5.** Uncropped immunoblot from *Figure 7c* (IP: EGFP; WB: Flag).

**Source data 6.** Uncropped immunoblot from *Figure 7c* (IP: EGFP; WB: EGFP).

**Source data 7.** Uncropped immunoblot from *Figure 7c* (Lysates, WB: Flag).

**Source data 8.** Uncropped immunoblot from *Figure 7e* (IP: EGFP; WB: CEP164).

**Source data 9.** Uncropped immunoblot from *Figure 7e* (IP: EGFP; WB: EGFP).

**Source data 10.** Uncropped immunoblot from *Figure 7e* (Lysates, WB: CEP164).

**Source data 11.** Uncropped immunoblot from *Figure 7g* (IP: EGFP; WB: Flag).

**Source data 12.** Uncropped immunoblot from *Figure 7g* (IP: EGFP; WB: EGFP).

**Source data 13.** Uncropped immunoblot from *Figure 7g* (Lysates, WB: Flag).

**Source data 14.** Source data from *Figure 7b*.

**Source data 15.** Source data from *Figure 7f*.

**Figure supplement 1.** CSNK2A1 regulates INPP5E ciliary targeting without strongly interacting with it.

**Figure supplement 1—source data 1.** Uncropped immunoblots from *Figure 7—figure supplement 1c*.

**Figure supplement 1—source data 2.** Uncropped immunoblot from *Figure 7—figure supplement 1c* (IP: EGFP; WB: Myc).

**Figure supplement 1—source data 3.** Uncropped immunoblot from *Figure 7—figure supplement 1c* (IP: EGFP; WB: EGFP).

**Figure supplement 1—source data 4.** Uncropped immunoblot from *Figure 7—figure supplement 1c* (Lysates; WB: Myc).

aa 1–467), middle (Mid, aa 468–1135) and C-terminal (CT, aa 1136–1460) regions (***Cajanek and Nigg, 2014***). Consistent with our hypothesis, CEP164(NT)-EGFP strongly interacted with Flag-INPP5E, as did full-length CEP164-EGFP (***Figure 7g***). Instead, no interaction was seen with CEP164(Mid)-EGFP and only a very weak one with CEP164(CT)-EGFP. Moreover, a mutation disrupting CEP164's WW domain (WWmut: Y73A+Y74 A) abolished INPP5E binding to CEP164(NT)-EGFP (***Figure 7g***; ***Cajanek and Nigg, 2014***). Hence, CEP164's NT is sufficient for INPP5E binding, and CEP164's WW domain is required for it.

Altogether, these data suggest a model of CLS2-3 action: presumably, excessively strong binding to CEP164 would retain INPP5E at the ciliary base and prevent its translocation into the ciliary compartment. CLS2-3 might overcome this by loosening the CEP164-INPP5E interaction. If this is the main

reason why CLS2-3 are required for INPP5E ciliary targeting, then deletion of the CEP164-interacting N-terminal region should rescue ciliary targeting of INPP5E-ΔCLS2 and INPP5E-ΔCLS3, as CLS2-3 would no longer be needed for CEP164 dissociation. To test this, we combined the ΔCLS2 and ΔCLS3 mutations with the Δ1–273 deletion, thereby generating the 274-644(ΔCLS2) and 274-644(ΔCLS3) mutants. Unlike the 274–644 control, which readily accumulated in cilia, both 274-644(ΔCLS2) and 274-644(ΔCLS3) completely failed to accumulate in cilia, just as the single ΔCLS2 and ΔCLS3 mutants (*Figure 7h*). Hence, even though CLS2-3 promote CEP164 dissociation, this is not sufficient for INPP5E ciliary targeting (*Figure 7i*). This might be due to CLS2-3 being required for binding to other ciliary trafficking proteins, such as TULP3 (*Figure 7a–d*).

The INPP5E-CEP164 interaction is also known to regulate ciliogenesis. Specifically, INPP5E prevents ectopic ciliogenesis by regulating the interaction between CEP164 and TTBK2, a key ciliogenic kinase that is in turn regulated by casein kinase 2 (CSNK2A1) (*Xu et al., 2016*; *Loukil et al., 2021*). Since both TTBK2-CEP164 and INPP5E-CEP164 interactions involve CEP164's WW domain, we reasoned that the CSNK2A1-TTBK2 pathway might also regulate INPP5E-CEP164 binding and hence INPP5E ciliary targeting (*Xu et al., 2016*). Indeed, we found INPP5E ciliary levels significantly reduced in CSNK2A1-null fibroblasts (*Figure 7—figure supplement 1a-b*). Also consistent with the above hypothesis, CSNK2A1 did not co-IP with INPP5E, suggesting that CSNK2A1's effect is indirect (*Figure 7—figure supplement 1c*). Whether it is mediated by TTBK2 awaits further study.

## INPP5E-ATG16L1 interaction is modulated by CLS1 and CLS4

Recent work shows that INPP5E ciliary targeting requires ATG16L1, an autophagy protein that forms a complex with IFT-B complex component IFT20 (*Boukhalfa et al., 2021*; *Finetti et al., 2021*). Moreover, ATG16L1 was shown to interact with INPP5E and its product PI4P (*Boukhalfa et al., 2021*). We therefore assessed the CLS-dependence of the ATG16L1-INPP5E interaction. Interestingly, although the single ΔCLS1, ΔCLS2, ΔCLS3, and ΔCLS4 mutants did not noticeably alter binding between EGFP-INPP5E and Flag-ATG16L1, a reduction was observed with the ΔCLS1 +4 mutant, suggesting that CLS1 and CLS4 may cooperate in ATG16L1 binding, thus mirroring their cooperation in INPP5E ciliary targeting (*Figure 8a*). Consistent with CLS1 and CLS4 partaking in the interaction, the C-terminal INPP5E fragment (251–644, containing both CLS1 and CLS4) was sufficient for binding, whereas the N-terminal fragment (1–283, containing only CLS1) interacted only weakly (*Figure 8a*). In three independent experiments, the ΔCLS1 +4 mutant was always found to interact less than the corresponding single mutants or WT (*Figure 8b*). However, despite the strong tendency, one-way ANOVA did not show a significant reduction with ΔCLS1 +4 (*Figure 8b*). Therefore, more independent experiments will be needed to clarify this point. In any case, these data suggest that CLS1 and CLS4 may jointly be implicated in how ATG16L1 targets INPP5E to cilia (*Figure 8c*). On the other hand, CLS2 and CLS3 would act via CEP164, TULP3, and ARL13B, with the latter also acting via CLS4, like PDE6D and RPGR (*Figure 8c*).

## INPP5E immune synapse targeting is CLS-independent

Most cell types form primary cilia when their centrosomes are not engaged in cell division. The main exception to this is the hematopoietic lineage, where the centrosome is often engaged in other specialized structures, such as the immune synapse (IS) in lymphocytes (*Stinchcombe and Griffiths, 2014*). Interestingly, numerous parallels exist between primary cilia and the IS, such as the use of IFT trafficking machinery (*Stinchcombe and Griffiths, 2014*; *Finetti et al., 2015*). ARL13B and ARL3 were shown to localize to the IS, and a recent preprint has shown that INPP5E does as well (*Powell et al., 2021*; *Stephen et al., 2018*; *Chiu et al., 2021*). With advice from the preprint authors, we confirmed that endogenous INPP5E accumulates at the IS between Jurkat T-cells and Raji antigen-presenting cells, a well-established IS model (*Figure 8—figure supplement 1a-b*; *Chiu et al., 2021*; *Herranz et al., 2019*; *Montoya et al., 2002*; *Calvo and Izquierdo, 2021*). Given the parallels between cilia and IS, we wondered whether IS targeting of INPP5E shares the same mechanisms that INPP5E uses for ciliary targeting in other cell types. To test this, we assessed the CLS-dependence of INPP5E IS targeting. As also reported in the preprint, EGFP-INPP5E WT is also detected at the Jurkat-Raji IS (*Figure 8—figure supplement 1c*; *Chiu et al., 2021*). This localization was not noticeably perturbed in the ΔCLS1, ΔCLS2, ΔCLS3, ΔCLS4, or ΔCLS1+4 mutants (*Figure 8—figure supplement 1c*). Therefore, INPP5E IS targeting does not follow the same mechanisms as INPP5E ciliary targeting.

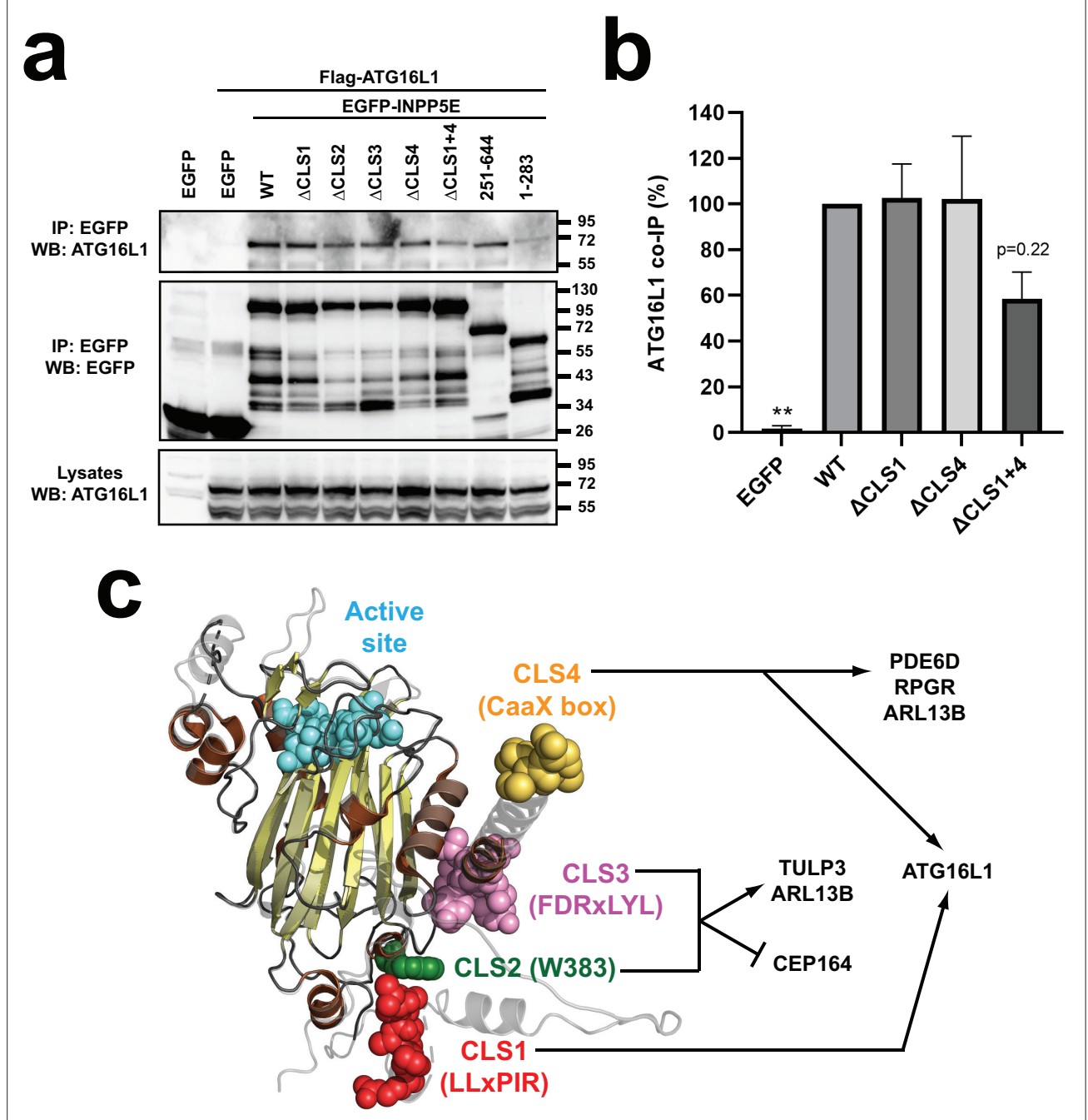

**Figure 8.** CLS1 and CLS4 jointly modulate the ATG16L1-INPP5E interaction. (**a**) The indicated EGFP-INPP5E constructs were coexpressed in HEK293T cells with Flag-ATG16L1, as shown. Lysates were immunoprecipitated with GFP-Trap beads and analyzed by Western blot with the indicated antibodies. (**b**) Quantitation of Flag-ATG16L1 co-IP by the indicated EGFP-INPP5E constructs in HEK293T cells. Co-IP levels, expressed as percentage of WT, are normalized by the amounts of both immunoprecipitated EGFP-INPP5Es and Flag-ATG16L1 lysate levels. Data are mean ± SEM of n=3 independent experiments and were analyzed by one-way ANOVA followed by Dunnett tests relative to WT. Significance is shown as p<0.01(**). (**c**) Schema of INPP5E structure depicting CLS1-4 and the proteins through which they regulate INPP5E ciliary targeting, as shown herein.

The online version of this article includes the following source data and figure supplement(s) for figure 8:

**Source data 1.** Uncropped immunoblots from *Figure 8a*.

**Source data 2.** Uncropped immunoblot from *Figure 8a* (IP: EGFP; WB: ATG16L1).

**Source data 3.** Uncropped immunoblot from *Figure 8a* (IP: EGFP; WB: EGFP).

**Source data 4.** Uncropped immunoblot from *Figure 8a* (Lysates, WB: ATG16L1).

*Figure 8 continued on next page*

*Figure 8 continued*

**Source data 5.** Source data from *Figure 8b*.

**Figure supplement 1.** INPP5E targeting to the T-cell immune synapse is CLS-independent.

## Discussion

In this work, we have demonstrated that INPP5E ciliary targeting is complex, requiring the interplay of four different CLSs. We have also made inroads into the mechanisms of action of these CLSs. *Table 1* summarizes our findings regarding how each CLS affects INPP5E's interactions with ciliary trafficking regulators PDE6D, RPGR, ARL13B, TULP3, CEP164, and ATG16L1.

Two of the CLSs (CLS2-3) are part of the catalytic domain, whose integrity is essential for ciliary localization. Moreover, mutations in CLS2-3 reduced enzyme activity (*Figure 2e*), raising the question of whether these are bona fide CLSs, actively engaging ciliary trafficking machinery, as opposed to simply maintaining overall domain architecture. We believe we have sufficiently demonstrated the former to be the case. Without being exhaustive, some observations proving this point include: (i) deleting CLS2 or CLS3 reduced activity and protein stability by about twofold, yet ciliary targeting was not merely reduced by half, but completely abolished (*Figure 2* and *Figure 2—figure supplements 2–4*); (ii) a similar twofold loss of activity and protein levels was seen for the R379A mutant, located directly adjacent to CLS2-3 in the catalytic domain, yet R379A was readily observed in cilia (*Figure 2* and *Figure 2—figure supplement 4*); and (iii) CLS2-3 specifically affect association of INPP5E to some ciliary trafficking regulators, but not others (*Figures 6–8*).

Together with our data showing CLS1 and CLS4 affect ciliary targeting without any effects on activity or protein stability (*Figure 3—figure supplements 1–2*), we feel confident that CLS1, 2, 3, and 4 are all bona fide CLSs. In support of this, all these CLSs are highly conserved, and they affect INPP5E function (*Figure 4*).

Although mutations in many other JBTS genes disrupt INPP5E ciliary localization, JBTS-causative mutations in the *INPP5E* gene itself had not been reported to do so (*Garcia-Gonzalo and Reiter, 2017*; *Bachmann-Gagescu et al., 2020*; *Garcia-Gonzalo et al., 2011*; *Roberson et al., 2015*; *Slaats et al., 2016*; *Alkanderi et al., 2018*; *Thomas et al., 2014*; *Humbert et al., 2012*; *Ning et al., 2021*; *Dowdle et al., 2011*). Instead, some of these *INPP5E* mutations, which invariably affect the catalytic domain, were shown to impair enzyme activity without affecting ciliary targeting (*Bielas et al., 2009*; *de Goede et al., 2016*). Here, we show for the first time that JBTS-causing *INPP5E* gene mutations do sometimes prevent ciliary accumulation (*Figure 5*). However, this does not appear to be due to specific inactivation of any CLS, as none of the mutations located near CLS2-3 prevented ciliary targeting. Rather, ciliary accumulation was lost in mutants likely to compromise catalytic domain integrity, such as G286R, V303M and W474R (*Figure 5*). Interestingly, both G286R and W474R are

**Table 1.** CLS-dependence of INPP5E protein-protein interactions.
Cilia localization and the indicated interactions are shown for each EGFP-INPP5E construct on the left column. For both localization and interactions, meaning of arrows is as follows: two upward green arrows (strong), one upward green arrow (moderate), one downward red arrow (low), and two downward red arrows (undetectable).

| INPP5E construct | Ciliary? | INPP5E interactors | | | | | |
| --- | --- | --- | --- | --- | --- | --- | --- |
| | | PDE6D | RPGR | ARL13B | TULP3 | CEP164 | ATG16L1 |
| WT | ↑↑ | ↑ | ↑ | ↑ | ↑ | ↑ | ↑ |
| ΔCLS1 | ↑ | ↑ | ↑ | ↑ | ↑ | ↑ | ↑ |
| ΔCLS2 | ↓↓ | ↑ | ↑ | ↑ | ↓ | ↑↑ | ↑ |
| ΔCLS3 | ↓↓ | ↑ | ↑ | ↑ | ↓ | ↑↑ | ↑ |
| ΔCLS4 | ↑ | ↓↓ | ↓↓ | ↓ | ↑ | ↑ | ↑ |
| ΔCLS1+4 | ↓↓ | ↓↓ | ↓↓ | ↓ | ↑ | ↑ | ↓ |
| CT (251-644) | ↑↑ | ↑ | ↑ | ↑ | ↑ | ↓↓ | ↑ |
| NT (1-283) | ↓↓ | ↓↓ | ↓↓ | ↓↓ | ↓ | ↑↑ | ↓ |

homozygous mutations, suggesting a complete lack of INPP5E in the cilia of these patients. Whether and how this contributed to JBTS manifestations in these patients, as compared to patients with activity-impaired but still ciliary INPP5E, remains an open question.

Besides discovering two novel CLSs in INPP5E (CLS1-2), we also shed light into how these novel CLSs relate to the previously identified ones (CLS3-4). Specifically, we found that CLS1 and CLS4 are partially redundant, with ciliary targeting moderately impaired in the single mutants, yet completely abolished in the double mutant (*Figure 3a–c*). Although the mechanisms of CLS4 action are fairly well understood, how CLS1 can partially substitute for it remains unclear (*Thomas et al., 2014*; *Gotthardt et al., 2015*; *Fansa et al., 2016*). In this regard, we tested several hypotheses, such as CLS1 also mediating binding to PDE6D, RPGR, or ARL13B, but none of this was the case (*Figure 6*; *Zhang et al., 2019*; *Rao et al., 2016*; *Dutta and Seo, 2016*; *Lee and Seo, 2015*; *Fansa et al., 2015*; *Wätzlich et al., 2013*; *Khanna, 2018*). ARL3 and BART are also involved in the CLS4 pathway, but we found that they do not co-IP with INPP5E (data not shown) (*Humbert et al., 2012*; *Gotthardt et al., 2015*; *Fansa et al., 2016*; *Ivanova et al., 2017*; *ElMaghloob et al., 2021*).

Partial redundancy between CLS1 and CLS4 might be partly explained by their effects on ATG16L1 binding (*Figure 8a–b*). ATG16L1 is an autophagy protein without which INPP5E cannot accumulate in cilia (*Boukhalfa et al., 2021*). As previously reported, we detected INPP5E-ATG16L1 binding, which appeared reduced with ΔCLS1 +4, but not ΔCLS1 or ΔCLS4. However, despite consistent results in three independent experiments and a strong statistical tendency in the ANOVA, the reduction in ΔCLS1 +4 was not statistically significant, so more experiments will be needed to clarify this point. Since ATG16L1 is involved in the Golgi exit of IFT20, an IFT-B component that traffics from Golgi to cilia base, one could speculate that ATG16L1 is needed for INPP5E Golgi exit (*Boukhalfa et al., 2021*). This would be consistent with previous reports of INPP5E Golgi localization, and with post-prenylation processing of CaaX box proteins occurring on the ER-Golgi surface (*Kong et al., 2000*; *Wright and Philips, 2006*). However, if ATG16L1 indeed works through CLS1 and CLS4, then this hypothesis predicts that INPP5E ΔCLS1 +4 will be retained at the Golgi. This is clearly not the case, though, as our data clearly show that ΔCLS1 +4 accumulates at the ciliary transition zone (*Figure 3* and *Figure 4—figure supplement 3*). Therefore, CLS1-4 exert their main functions at the transition zone, and how ATG16L1 mediates INPP5E ciliary targeting remains poorly understood.

Besides the functional CLS1-CLS4 connection, we also uncovered a CLS2-CLS3 link. Given their steric proximity (*Figure 2b*), we hypothesize CLS2 and CLS3 work together as a functional unit to recruit ciliary trafficking proteins like ARL13B and TULP3 (*Figure 8c*). Consistent with this, CLS2 and CLS3 behaved largely equivalently in all our interaction experiments (*Figures 6–8*). Moreover, the need to keep CLS2-3 together also explains why a folded catalytic domain is critical for ciliary targeting (*Figure 1*). Future structural studies should further clarify these issues.

ARL13B regulates INPP5E ciliary targeting by functioning as an ARL3-GEF, thereby promoting INPP5E release from PDE6D inside cilia (*Thomas et al., 2014*; *Humbert et al., 2012*; *Fujisawa et al., 2021*; *Qiu et al., 2021*; *Gotthardt et al., 2015*). However, this indirect connection does not suffice to explain why CLS4 promotes association between ARL13B and INPP5E, as we and others have found (*Figure 6*; *Fujisawa et al., 2021*). The explanation for this might be that, by mediating INPP5E membrane insertion, CLS4-dependent farnesylation facilitates access to ARL13B, a fatty acylated protein (*Cevik et al., 2013*; *Roy et al., 2017*). If so, an alternative membrane anchor in ΔCLS4 should restore the interaction. ARL13B has also been shown to directly interact with INPP5E in a CLS3-dependent manner, an interaction that is essential for ARL13B to mediate INPP5E ciliary retention (*Fujisawa et al., 2021*; *Qiu et al., 2021*). Although CLS2-3 mutants did not affect INPP5E-ARL13B co-IP in our conditions, our co-recruitment assays did show a strong effect of both CLS2 and CLS3 on this interaction. Given that CLS2-3 likely form a functional unit as mentioned above, CLS2 probably affects the direct ARL13B-INPP5E interaction, as previously reported for CLS3 (*Humbert et al., 2012*).

TULP3 is essential for ciliary targeting of both ARL13B and INPP5E, among other ciliary membrane proteins (*Han et al., 2019*; *Palicharla et al., 2021*). Through its C-terminal Tubby domain, TULP3 directly interacts with ARL13B, targeting it to cilia, which in turn allows INPP5E ciliary targeting (*Palicharla et al., 2021*). However, whether TULP3 and INPP5E interact with each other had not been reported. Our co-IPs show that they do, in a CLS2-3 and Tubby domain-dependent manner (*Figure 7a–c*). Tubby domain's mutations K389A and K268A+R270 A reduced but not abolished the interaction. K389 is important for direct TULP3-ARL13B binding, and critical for ciliary targeting of

both ARL13B and INPP5E (*Palicharla et al., 2021*). This suggests the TULP3-INPP5E interaction might be mediated by ARL13B. However, if INPP5E associates to TULP3 indirectly via ARL13B, then it is hard to explain why ΔCLS4 reduces INPP5E-ARL13B but not INPP5E-TULP3 co-IP, and why ΔCLS2-3 do the opposite (*Figures 6–7*). Thus, INPP5E-TULP3 probably do not interact via ARL13B. Whether or not they interact directly remains to be determined. On the other hand, the K268A+R270 A mutation, which interferes with TULP3's ability to bind phosphoinositides and target transmembrane receptors to cilia, has no effect on TULP3's ability to target ARL13B and INPP5E to cilia (*Palicharla et al., 2021*). Whether this mutation affects other functional aspects of the TULP3-INPP5E interaction we observe remains an open possibility.

CEP164 binds INPP5E and recruits it to the ciliary base (*Humbert et al., 2012*). Herein, we show that this interaction involves the N-termini of both proteins, with the WW domain in CEP164 being critical (*Figure 7e–i*). Since WW domains typically interact with proline-rich motifs, which abound in INPP5E's N-terminal region, it seems likely that the CEP164-INPP5E interaction also involves these motifs (*Macias et al., 2002*). We also provide evidence that this interaction is negatively regulated by CLS2-3 (*Figure 7e–f*). This is interesting, as the CEP164-INPP5E interaction may need to be loosened for INPP5E to efficiently enter cilia. Such loosening, however, cannot be the main function of CLS2-3, as they were still required for ciliary targeting of INPP5E mutants unable to bind CEP164 (*Figure 7h*). Still, even if the loosening up of the CEP164-INPP5E interaction is not sufficient, it may still be necessary for INPP5E ciliary targeting. This remains to be determined. If so, one possible mechanism would be competition between Tau tubulin kinase 2 (TTBK2) and INPP5E for CEP164's WW domain, which they both bind through proline-rich regions (*Xu et al., 2016*). If so, this could explain our observation that casein kinase CSNK2A1, a recently reported upstream regulator of TTBK2, promotes INPP5E ciliary targeting without interacting with it (*Figure 7—figure supplement 1*; *Loukil et al., 2021*).

Finally, given the known parallels between primary cilia and immune synapses, with the latter containing, among other ciliary proteins, ARL13B and INPP5E, we asked whether CLS1-4 also drive INPP5E targeting to the IS (*Stephen et al., 2018*; *Chiu et al., 2021*). The answer was no, pointing to clearly distinct mechanisms for targeting to both structures (*Figure 7—figure supplement 1*). This may mean that INPP5E targeting to cilia and IS evolved independently of each other or, alternatively, that a common evolutionary origin has been blurred by a long history of evolutionary divergence. The fact that INPP5E IS targeting is quickly induced upon assembly of the highly dynamic IS also suggests a more transient role for INPP5E at the IS, as opposed to its more constitutive ciliary localization. The mechanisms of INPP5E targeting to the IS await further investigation.

Altogether, our data show that INPP5E ciliary targeting is a surprisingly complex process involving four different cis-acting sequences (CLS1-4), and multiple trans-acting factors (like PDE6D, RPGR, ARL13B, TULP3, CEP164, and ATG16L1). This level of complexity is unusual, especially when compared to other ciliary cargoes, whose targeting typically involves a single CLS (*Nachury and Mick, 2019*; *Garcia-Gonzalo and Reiter, 2012*; *Barbeito and Garcia-Gonzalo, 2021*; *Mukhopadhyay et al., 2017*; *McIntyre et al., 2016*). The complexity and redundancy in INPP5E ciliary targeting suggest this is an important process, subject to fine regulation. This is consistent with the surprisingly wide range of functions INPP5E plays at the cilium, controlling among others their lipid and protein composition, assembly and disassembly, exovesicle release, and signaling (*Jacoby et al., 2009*; *Bielas et al., 2009*; *Guo et al., 2019*; *Garcia-Gonzalo et al., 2015*; *Chávez et al., 2015*; *Badgandi et al., 2017*; *Dyson et al., 2017*; *Phua et al., 2017*; *Wang et al., 2011*; *Hakim et al., 2016*; *Xu et al., 2016*; *Xu et al., 2017*; *Constable et al., 2020*; *Hasenpusch-Theil et al., 2020*; *Sharif et al., 2021*; *Ukhanov et al., 2022*; *Yue et al., 2021*). With the broader view of INPP5E ciliary targeting provided herein, the stage is now set for a deeper molecular understanding of these processes and their regulation.

## Materials and methods

**Key resources table**

| Reagent type (species) or resource | Designation | Source or reference | Identifiers | Additional information |
|---|---|---|---|---|
| Cell line (*Homo sapiens*) | hTERT-RPE1 | ATCC | Cat # CRL-4000 | Derived from retinal pigmented epithelium |

*Continued on next page*

*Continued*

| Reagent type (species) or resource | Designation | Source or reference | Identifiers | Additional information |
|---|---|---|---|---|
| Cell line (*Homo sapiens*) | Puromycin-sensitive hTERT-RPE1 | This study | Puromycin Acetyltransferase (PAC)-KO cells | Also used in *Gonçalves et al., 2021* |
| Cell line (*Homo sapiens*) | INPP5E-KO hTERT-RPE1 | This study | Clone 3 | Derived from puromycin-sensitive hTERT-RPE1 |
| Cell line (*Homo sapiens*) | INPP5E-KO hTERT-RPE1 | This study | Clone 12 | Derived from puromycin-sensitive hTERT-RPE1 |
| Cell line (*Homo sapiens*) | 293T | ATCC | Cat # CRL-3216 | Derived from human embryonic kidney |
| Cell line (*Homo sapiens*) | HeLa | ATCC | Cat # CCL-2 | Derived from cervical carcinoma |
| Cell line (*Homo sapiens*) | Jurkat, Clone E6-1 | ATCC | Cat # TIB-152 | T lymphoblasts from acute T cell leukemia |
| Cell line (*Homo sapiens*) | Raji | ATCC | Cat # CCL-86 | Lymphoblast-like cells from Burkitt's lymphoma |
| Cell line (*Mus musculus*) | Csnk2a1-WT MEFs | *Loukil et al., 2021* | N/A | Control mouse embryonic fibroblasts (MEFs) |
| Cell line (*Mus musculus*) | Csnk2a1-KO MEFs | *Loukil et al., 2021* | N/A | Casein kinase 2 subunit alpha-null MEFs |
| Antibody | Anti-acetylated α-tubulin (mouse monoclonal) | Sigma-Aldrich (Merck) | Cat # T7451 (clone 6-11B-1) | IF: 1:10,000 |
| Antibody | Anti-α-tubulin (mouse monoclonal) | Proteintech | Cat # 66031–1-Ig | WB: 1:1000 |
| Antibody | Anti-γ-tubulin (mouse monoclonal) | Santa Cruz | Cat # sc-17787 | IF: 1:200 |
| Antibody | Anti-EGFP (mouse monoclonal) | Proteintech | Cat # 66002–1-Ig | WB: 1:1000 |
| Antibody | Anti-EGFP (mouse monoclonal) | Santa Cruz | Cat # sc-9996 | IF: 1:200 |
| Antibody | Anti-ARL13B (mouse monoclonal) | Proteintech | Cat # 66739–1-Ig | WB: 1:1000 |
| Antibody | Anti-Flag (mouse monoclonal) | Sigma-Aldrich (Merck) | Cat # F3165 (clone M2) | WB: 1:2000 |
| Antibody | Anti-polyglutamylated tubulin (mouse monoclonal) | Adipogen | Cat # GT335 | IF: 1:2000 |
| Antibody | Anti-EGFP (rabbit polyclonal) | Proteintech | Cat # 50430–2-AP | IF: 1:200 WB: 1:1000 |
| Antibody | Anti-RPGR (rabbit polyclonal) | Proteintech | Cat # 16891–1-AP | WB: 1:1000 |
| Antibody | Anti-TULP3 (rabbit polyclonal) | Proteintech | Cat # 13637–1-AP | IF: 1:750 WB: 1:2000 |
| Antibody | Anti-CEP164 (rabbit polyclonal) | Proteintech | Cat # 22227–1-AP | WB: 1:1000 |
| Antibody | Anti-ATG16L (rabbit polyclonal) | MBL | Cat # PM040 | WB: 1:1000 |

*Continued*

| Reagent type (species) or resource | Designation | Source or reference | Identifiers | Additional information |
|---|---|---|---|---|
| Antibody | Anti-Myc (rabbit polyclonal) | Proteintech | Cat # 16286–1-AP | WB: 1:1000 |
| Antibody | Anti-INPP5E (rabbit polyclonal) | Proteintech | Cat # 17797–1-AP | IF: 1:100 |
| Antibody | Anti- γ-tubulin (goat polyclonal) | Santa Cruz | Cat # sc-7396 (discontinued) | IF: 1:200 |
| Antibody | Alexa Fluor 488 donkey anti-rabbit IgG (donkey polyclonal) | Thermo Fisher | Cat # A21206 | IF: 1:10,000 |
| Antibody | Alexa Fluor 555 donkey anti-mouse IgG (donkey polyclonal) | Thermo Fisher | Cat # A31570 | IF: 1:10,000 |
| Antibody | Alexa Fluor 647 donkey anti-goat IgG (donkey polyclonal) | Thermo Fisher | Cat # A21447 | IF: 1:10,000 |
| Antibody | Alexa Fluor 488 goat anti-mouse IgG2a (goat polyclonal) | Thermo Fisher | Cat # A21131 | IF: 1:10,000 |
| Antibody | Alexa Fluor 555 goat anti-mouse IgG1 (goat polyclonal) | Thermo Fisher | Cat # A21127 | IF: 1:10,000 |
| Antibody | Alexa Fluor 555 goat anti-mouse IgG2b (goat polyclonal) | Thermo Fisher | Cat # A21147 | IF: 1:10,000 |
| Antibody | Alexa Fluor 647 goat anti-mouse IgG2a (goat polyclonal) | Thermo Fisher | Cat # A21241 | IF: 1:10,000 |
| Antibody | HRP-conjugated goat anti-mouse IgG (goat polyclonal) | Thermo Fisher | Cat # A16072 | WB: 62 ng/ml |
| Antibody | HRP-conjugated goat anti-rabbit IgG (goat polyclonal) | Thermo Fisher | Cat # A16104 | WB: 62 ng/ml |
| Antibody | GFP-Trap_MA magnetic agarose beads (alpaca monoclonal) | Chromotek (Proteintech) | Cat # gtma-20 | IP: 10 µl slurry for 500 µl lysate (1:50) |
| Recombinant DNA reagent | EGFP-INPP5E | *Jacoby et al., 2009* | Human INPP5E NM_019892.6 (644 amino acids) | XhoI-KpnI cloning into pEGFP-C1 |
| Recombinant DNA reagent | EGFP-INPP5E (MORM) | *Jacoby et al., 2009* | 1–626 (MORM: Δ627–644) | XhoI-KpnI cloning into pEGFP-C1 |
| Recombinant DNA reagent | EGFP-INPP5E (D477N) | *Garcia-Gonzalo et al., 2015* | D477N | XhoI-KpnI cloning into pEGFP-C1 |
| Recombinant DNA reagent | EGFP-INPP5E (1-623) | This study | 1–623 | XhoI-KpnI cloning into pEGFP-C1 |
| Recombinant DNA reagent | EGFP-INPP5E (1-621) | This study | 1–621 | XhoI-KpnI cloning into pEGFP-C1 |
| Recombinant DNA reagent | EGFP-INPP5E (1-618) | This study | 1–618 | XhoI-KpnI cloning into pEGFP-C1 |

*Continued*

| Reagent type (species) or resource | Designation | Source or reference | Identifiers | Additional information |
|---|---|---|---|---|
| Recombinant DNA reagent | EGFP-INPP5E (1-616) | This study | 1–616 | XhoI-KpnI cloning into pEGFP-C1 |
| Recombinant DNA reagent | EGFP-INPP5E (1-608) | This study | 1–608 | XhoI-KpnI cloning into pEGFP-C1 |
| Recombinant DNA reagent | EGFP-INPP5E (1-283) | This study | 1–283 | XhoI-KpnI cloning into pEGFP-C1 |
| Recombinant DNA reagent | EGFP-INPP5E (Δ297–599) | This study | Δ297–599 | XhoI-KpnI cloning into pEGFP-C1 |
| Recombinant DNA reagent | EGFP-INPP5E (100-644) | This study | 100–644 | XhoI-KpnI cloning into pEGFP-C1 |
| Recombinant DNA reagent | EGFP-INPP5E (200-644) | This study | 200–644 | XhoI-KpnI cloning into pEGFP-C1 |
| Recombinant DNA reagent | EGFP-INPP5E (251-644) | This study | 251–644 | XhoI-KpnI cloning into pEGFP-C1 |
| Recombinant DNA reagent | EGFP-INPP5E (274-644) | This study | 274–644 | XhoI-KpnI cloning into pEGFP-C1 |
| Recombinant DNA reagent | EGFP-INPP5E (288-644) | This study | 288–644 | XhoI-KpnI cloning into pEGFP-C1 |
| Recombinant DNA reagent | EGFP-INPP5E (351-644) | This study | 351–644 | XhoI-KpnI cloning into pEGFP-C1 |
| Recombinant DNA reagent | EGFP-INPP5E (451-644) | This study | 451–644 | XhoI-KpnI cloning into pEGFP-C1 |
| Recombinant DNA reagent | EGFP-INPP5E (551-644) | This study | 551–644 | XhoI-KpnI cloning into pEGFP-C1 |
| Recombinant DNA reagent | EGFP-INPP5E (FDR609AAA) | This study | F609A+D610A+R611 A (aka ΔCLS3 or FDR >AAA) | XhoI-KpnI cloning into pEGFP-C1 |
| Recombinant DNA reagent | EGFP-INPP5E (LYL613AAA) | This study | L613A+Y614A+L615 A (aka LYL >AAA) | XhoI-KpnI cloning into pEGFP-C1 |
| Recombinant DNA reagent | EGFP-INPP5E (F609A) | This study | F609A | XhoI-KpnI cloning into pEGFP-C1 |
| Recombinant DNA reagent | EGFP-INPP5E (D610A) | This study | D610A | XhoI-KpnI cloning into pEGFP-C1 |
| Recombinant DNA reagent | EGFP-INPP5E (R611A) | This study | R611A | XhoI-KpnI cloning into pEGFP-C1 |
| Recombinant DNA reagent | EGFP-INPP5E (E612A) | This study | E612A | XhoI-KpnI cloning into pEGFP-C1 |
| Recombinant DNA reagent | EGFP-INPP5E (L613A) | This study | L613A | XhoI-KpnI cloning into pEGFP-C1 |
| Recombinant DNA reagent | EGFP-INPP5E (Y614A) | This study | Y614A | XhoI-KpnI cloning into pEGFP-C1 |
| Recombinant DNA reagent | EGFP-INPP5E (L615A) | This study | L615A | XhoI-KpnI cloning into pEGFP-C1 |
| Recombinant DNA reagent | EGFP-INPP5E (R345A+R346 A) | This study | R345A+R346 A | XhoI-KpnI cloning into pEGFP-C1 |
| Recombinant DNA reagent | EGFP-INPP5E (E347A) | This study | E347A | XhoI-KpnI cloning into pEGFP-C1 |
| Recombinant DNA reagent | EGFP-INPP5E (W348A) | This study | W348A | XhoI-KpnI cloning into pEGFP-C1 |

*Continued on next page*

*Continued*

| Reagent type (species) or resource | Designation | Source or reference | Identifiers | Additional information |
|---|---|---|---|---|
| Recombinant DNA reagent | EGFP-INPP5E (E349A) | This study | E349A | XhoI-KpnI cloning into pEGFP-C1 |
| Recombinant DNA reagent | EGFP-INPP5E (Q353A) | This study | Q353A | XhoI-KpnI cloning into pEGFP-C1 |
| Recombinant DNA reagent | EGFP-INPP5E (E354A) | This study | E354A | XhoI-KpnI cloning into pEGFP-C1 |
| Recombinant DNA reagent | EGFP-INPP5E (Y360A) | This study | Y360A | XhoI-KpnI cloning into pEGFP-C1 |
| Recombinant DNA reagent | EGFP-INPP5E (Y360F) | This study | Y360F | XhoI-KpnI cloning into pEGFP-C1 |
| Recombinant DNA reagent | EGFP-INPP5E (V361A) | This study | V361A | XhoI-KpnI cloning into pEGFP-C1 |
| Recombinant DNA reagent | EGFP-INPP5E (R378A+R379 A) | This study | R378A+R379 A | XhoI-KpnI cloning into pEGFP-C1 |
| Recombinant DNA reagent | EGFP-INPP5E (R378A) | This study | R378A | XhoI-KpnI cloning into pEGFP-C1 |
| Recombinant DNA reagent | EGFP-INPP5E (R379A) | This study | R379A | XhoI-KpnI cloning into pEGFP-C1 |
| Recombinant DNA reagent | EGFP-INPP5E (D380A) | This study | D380A | XhoI-KpnI cloning into pEGFP-C1 |
| Recombinant DNA reagent | EGFP-INPP5E (I382A) | This study | I382A | XhoI-KpnI cloning into pEGFP-C1 |
| Recombinant DNA reagent | EGFP-INPP5E (W383A) | This study | W383A (aka ΔCLS2) | XhoI-KpnI cloning into pEGFP-C1 |
| Recombinant DNA reagent | EGFP-INPP5E (W383F) | This study | W383F | XhoI-KpnI cloning into pEGFP-C1 |
| Recombinant DNA reagent | EGFP-INPP5E (W383I) | This study | W383I | XhoI-KpnI cloning into pEGFP-C1 |
| Recombinant DNA reagent | EGFP-INPP5E (W383L) | This study | W383L | XhoI-KpnI cloning into pEGFP-C1 |
| Recombinant DNA reagent | EGFP-INPP5E (W383M) | This study | W383M | XhoI-KpnI cloning into pEGFP-C1 |
| Recombinant DNA reagent | EGFP-INPP5E (W383V) | This study | W383V | XhoI-KpnI cloning into pEGFP-C1 |
| Recombinant DNA reagent | EGFP-INPP5E (W383E) | This study | W383E | XhoI-KpnI cloning into pEGFP-C1 |
| Recombinant DNA reagent | EGFP-INPP5E (W383R) | This study | W383R | XhoI-KpnI cloning into pEGFP-C1 |
| Recombinant DNA reagent | EGFP-INPP5E (F384A) | This study | F384A | XhoI-KpnI cloning into pEGFP-C1 |
| Recombinant DNA reagent | EGFP-INPP5E (E387A) | This study | E387A | XhoI-KpnI cloning into pEGFP-C1 |
| Recombinant DNA reagent | EGFP-INPP5E (Δ251–273) | This study | Δ251–273 (aka ΔCLS1) | XhoI-KpnI cloning into pEGFP-C1 |
| Recombinant DNA reagent | EGFP-INPP5E (C641S) | This study | C641S (aka ΔCLS4) | XhoI-KpnI cloning into pEGFP-C1 |
| Recombinant DNA reagent | EGFP-INPP5E (Δ251–273+C641 S) | This study | Δ251–273+C641 S (aka ΔCLS1+4) | XhoI-KpnI cloning into pEGFP-C1 |

*Continued*

| Reagent type (species) or resource | Designation | Source or reference | Identifiers | Additional information |
|---|---|---|---|---|
| Recombinant DNA reagent | EGFP-INPP5E (288-626) | This study | 288–626 | XhoI-KpnI cloning into pEGFP-C1 |
| Recombinant DNA reagent | EGFP-INPP5E (274-626) | This study | 274–626 | XhoI-KpnI cloning into pEGFP-C1 |
| Recombinant DNA reagent | EGFP-INPP5E (269-626) | This study | 269–626 | XhoI-KpnI cloning into pEGFP-C1 |
| Recombinant DNA reagent | EGFP-INPP5E (264-626) | This study | 264–626 | XhoI-KpnI cloning into pEGFP-C1 |
| Recombinant DNA reagent | EGFP-INPP5E (257-626) | This study | 257–626 | XhoI-KpnI cloning into pEGFP-C1 |
| Recombinant DNA reagent | EGFP-INPP5E (251-626) | This study | 251–626 | XhoI-KpnI cloning into pEGFP-C1 |
| Recombinant DNA reagent | EGFP-INPP5E (257-626) +FS257AA | This study | (257-626)+F257A+S258 A | XhoI-KpnI cloning into pEGFP-C1 |
| Recombinant DNA reagent | EGFP-INPP5E (257-626) +LL259AA | This study | (257-626)+L259A+L260 A | XhoI-KpnI cloning into pEGFP-C1 |
| Recombinant DNA reagent | EGFP-INPP5E (257-626) +PIR262AAA | This study | (257-626)+P262A+I263A+R264 A | XhoI-KpnI cloning into pEGFP-C1 |
| Recombinant DNA reagent | EGFP-INPP5E (257-626) +SK265AA | This study | (257-626)+S265A+K266 A | XhoI-KpnI cloning into pEGFP-C1 |
| Recombinant DNA reagent | EGFP-INPP5E (257-626) +DV267AA | This study | (257-626)+D267A+V268 A | XhoI-KpnI cloning into pEGFP-C1 |
| Recombinant DNA reagent | EGFP-INPP5E (257-626) +L259 A | This study | (257-626)+L259 A | XhoI-KpnI cloning into pEGFP-C1 |
| Recombinant DNA reagent | EGFP-INPP5E (257-626) +L260 A | This study | (257-626)+L260 A | XhoI-KpnI cloning into pEGFP-C1 |
| Recombinant DNA reagent | EGFP-INPP5E (257-626) +P262 A | This study | (257-626)+P262 A | XhoI-KpnI cloning into pEGFP-C1 |
| Recombinant DNA reagent | EGFP-INPP5E (257-626) +I263 A | This study | (257-626)+I263 A | XhoI-KpnI cloning into pEGFP-C1 |
| Recombinant DNA reagent | EGFP-INPP5E (257-626) +R264 A | This study | (257-626)+R264 A | XhoI-KpnI cloning into pEGFP-C1 |
| Recombinant DNA reagent | EGFP-INPP5E (G286R) | This study | G286R | XhoI-KpnI cloning into pEGFP-C1 |
| Recombinant DNA reagent | EGFP-INPP5E (V303M) | This study | V303M | XhoI-KpnI cloning into pEGFP-C1 |
| Recombinant DNA reagent | EGFP-INPP5E (R345S) | This study | R345S | XhoI-KpnI cloning into pEGFP-C1 |
| Recombinant DNA reagent | EGFP-INPP5E (T355M) | This study | T355M | XhoI-KpnI cloning into pEGFP-C1 |

*Continued on next page*

*Continued*

| Reagent type (species) or resource | Designation | Source or reference | Identifiers | Additional information |
|---|---|---|---|---|
| Recombinant DNA reagent | EGFP-INPP5E (R378C) | This study | R378C | XhoI-KpnI cloning into pEGFP-C1 |
| Recombinant DNA reagent | EGFP-INPP5E (C385Y) | This study | C385Y | XhoI-KpnI cloning into pEGFP-C1 |
| Recombinant DNA reagent | EGFP-INPP5E (V388L) | This study | V388L | XhoI-KpnI cloning into pEGFP-C1 |
| Recombinant DNA reagent | EGFP-INPP5E (W474R) | This study | W474R | XhoI-KpnI cloning into pEGFP-C1 |
| Recombinant DNA reagent | EGFP-INPP5E (D490Y) | This study | D490Y | XhoI-KpnI cloning into pEGFP-C1 |
| Recombinant DNA reagent | EGFP-INPP5E (R621Q) | This study | R621Q | XhoI-KpnI cloning into pEGFP-C1 |
| Recombinant DNA reagent | EGFP-INPP5E (R621W) | This study | R621W | XhoI-KpnI cloning into pEGFP-C1 |
| Recombinant DNA reagent | EGFP-INPP5E (C641R) | This study | C641R | XhoI-KpnI cloning into pEGFP-C1 |
| Recombinant DNA reagent | EGFP-INPP5E (274-644) + ΔCLS2 | This study | (274-644)+W383 A | XhoI-KpnI cloning into pEGFP-C1 |
| Recombinant DNA reagent | EGFP-INPP5E (274-644) + ΔCLS3 | This study | (274-644)+F609A+D610A+R611 A | XhoI-KpnI cloning into pEGFP-C1 |
| Recombinant DNA reagent | Flag-INPP5E | This study | Human INPP5E NM_019892.6 (644 amino acids) | EcoRI-KpnI cloning into pFlag-CMV4 |
| Recombinant DNA reagent | mVenus-PDE6D | This study | Human PDE6D NM_002601.4 (150 amino acids) | Cloned into p-mVenus-C1 |
| Recombinant DNA reagent | Flag-PDE6D | This study | Human PDE6D NM_002601.4 (150 amino acids) | mVenus-PDE6D cassette from eponymous plasmid excised with AgeI-EcoRI and replaced by Flag-PDE6D |
| Recombinant DNA reagent | pcDNA3.1(+)-N-DYK-RPGR | GenScript | Human RPGR NM_000328.3 (815 amino acids) | Plasmid, expresses Flag-RPGR (aka DYK-RPGR) |
| Recombinant DNA reagent | ARL13B-EYFP | This study | Human ARL13B NM_001174150.2 (428 amino acids) | Cloned into pEYFP-C1 |
| Recombinant DNA reagent | ARL13B-EGFP | This study | Human ARL13B NM_001174150.2 (428 amino acids) | XhoI-BamHI cloning into pEGFP-N1 |
| Recombinant DNA reagent | ARL13B-Flag | This study | Human ARL13B NM_001174150.2 (428 amino acids) | EGFP in ARL13B-EGFP was swapped by Flag using AgeI-NotI and pre-annealed Flag-encoding primers. |
| Recombinant DNA reagent | pcDNA3.1-TULP3-myc-his | *Barbeito and Garcia-Gonzalo, 2021* | Human TULP3 NP_003315.2 (442 amino acids) | XhoI-BamHI into pcDNA3.1-myc-his(-)C |
| Recombinant DNA reagent | pG-LAP1-TULP3 | *Mukhopadhyay et al., 2010* | Human TULP3 NP_003315.2 (442 amino acids) | Expresses EGFP-Stag-TULP3 (LAP-TULP3) |
| Recombinant DNA reagent | pG-LAP1-TULP3-KR | *Mukhopadhyay et al., 2010* | TULP3 (K268A+R270 A) | Phosphoinositide binding-defective mutant |
| Recombinant DNA reagent | EGFP-TULP3 (NTD) | This study | TULP3 (1–183) | KpnI-BamHI cloning into pEGFP-C1 |

*Continued on next page*

*Continued*

| Reagent type (species) or resource | Designation | Source or reference | Identifiers | Additional information |
|---|---|---|---|---|
| Recombinant DNA reagent | EGFP-TULP3 (CTD) | This study | TULP3 (184–442) (aka Tubby domain) | KpnI-BamHI cloning into pEGFP-C1 |
| Recombinant DNA reagent | EGFP-TULP3 (K389A) | This study | Human TULP3 NP_003315.2 (442 amino acids) | KpnI-BamHI cloning into pEGFP-C1 |
| Recombinant DNA reagent | CEP164-EGFP | Addgene | Addgene plasmid; RRID:Addgene_41149 | Human CEP164 NM_014956.5 (1460 amino acids) |
| Recombinant DNA reagent | CEP164-EGFP (1-467) | This study | CEP164-NT (aka N-term in *Cajanek and Nigg, 2014*) | CEP164-NT replaces full length in CEP164-EGFP (EcoRI-KpnI) |
| Recombinant DNA reagent | CEP164-EGFP (1-467)+WWmut | This study | (1-467) +Y73A+Y74 A | Mutant of WW domain |
| Recombinant DNA reagent | CEP164-EGFP (468–1135) | This study | CEP164-Mid (aka M-part in *Cajanek and Nigg, 2014*) | CEP164-Mid replaces full length in CEP164-EGFP (EcoRI-KpnI) |
| Recombinant DNA reagent | CEP164-EGFP (1136–1460) | This study | CEP164-CT (aka C-term in *Cajanek and Nigg, 2014*) | CEP164-CT replaces full length in CEP164-EGFP (EcoRI-KpnI) |
| Recombinant DNA reagent | pMRX-IP-SECFP-hATG16A1 | Addgene | Addgene plasmid; RRID:Addgene_58994 | Human ATG16L1 NP_060444.3 (588 amino acids) |
| Recombinant DNA reagent | Flag-ATG16L1 | This study | Human ATG16L1 NP_060444.3 (588 amino acids) | EcoRI insert from pMRX-IP-SECFP-hATG16A1 transferred to pFlagCMV4 |
| Recombinant DNA reagent | CSNK2A1-myc | This study | Mouse CSNK2A1 NP_031814.2 (391 amino acids) | XhoI-BamHI into pcDNA3.1-myc-his(-)C |
| Recombinant DNA reagent | FRB-CFP-CaaX | *Roy et al., 2020* Biorxiv | FRB* domain of human mTOR (as in RRID:Addgene_20148) | CaaX box of K-Ras targets FRB-CFP to inner leaflet of plasma membrane |
| Recombinant DNA reagent | mCherry-FKBP | This study | Human FKBP1A (aka FKBP12) NP_000792.1 (108 amino acids) | Cloned into p-mCherry-C1 |
| Recombinant DNA reagent | mCherry-FKBP-INPP5E(WT) | This study | Human INPP5E NM_019892.6 (644 amino acids) | Cloned into mCherry-FKBP |
| Recombinant DNA reagent | mCherry-FKBP-INPP5E(ΔCLS1) | This study | INPP5E Δ251–273 | Cloned into mCherry-FKBP |
| Recombinant DNA reagent | mCherry-FKBP-INPP5E(ΔCLS2) | This study | INPP5E W383A | Cloned into mCherry-FKBP |
| Recombinant DNA reagent | mCherry-FKBP-INPP5E(ΔCLS3) | This study | INPP5E F609A+D610A+R611 A | Cloned into mCherry-FKBP |
| Recombinant DNA reagent | mCherry-FKBP-INPP5E(ΔCLS4) | This study | INPP5E C641S | Cloned into mCherry-FKBP |
| Recombinant DNA reagent | mCherry-FKBP-INPP5E (ΔCLS1+4) | This study | INPP5E Δ251–273+C641 S | Cloned into mCherry-FKBP |
| Recombinant DNA reagent | pSpCas9-sgPAC1 | This study | PAC gRNA#1 plasmid | PAC gRNA#1: ACGCGCGUCGGGCTCG ACAUCGG |
| Recombinant DNA reagent | pSpCas9-sgPAC3 | This study | PAC gRNA#3 plasmid | PAC gRNA#3: CACGCGCCACACCGUC GAUCCGG |

*Continued on next page*

*Continued*

| Reagent type (species) or resource | Designation | Source or reference | Identifiers | Additional information |
|---|---|---|---|---|
| Recombinant DNA reagent | pSpCas9-sgPAC6 | This study | PAC gRNA#3 plasmid | PAC gRNA#6: GGCGGGGUAGUCGGCG AACGCGG |
| Recombinant DNA reagent | pSpCas9-hINPP5E-gRNA1 | This study | INPP5E gRNA#1 plasmid | INPP5E gRNA#1: CGGAGCCCGGAGCAUC GGGUGGG |
| Recombinant DNA reagent | pSpCas9-hINPP5E-gRNA2 | This study | INPP5E gRNA#2 plasmid | INPP5E gRNA#2: UGGAGCGUCCUCCCUU CCGGCGG |
| Recombinant DNA reagent | pSpCas9-hINPP5E-gRNA3 | This study | INPP5E gRNA#3 plasmid | INPP5E gRNA#3: ACAGCUUCCCGGCGCUCCGCCGG |
| Commercial assay or kit | Malachite Green Assay Kit | Echelon Biosciences (Tebu-Bio) | Cat # K-1500 | For measurement of phosphate release in activity assays. |
| Chemical compound, drug | PtdIns(4,5)P2-diC8 | Echelon Biosciences (Tebu-Bio) | Cat # P-4508 | Activity assays: 120 µM |
| Chemical compound, drug | n-octyl-β-D-glucopyranoside | Alfa Aesar | Cat # J67390.03 | Activity assays: 0.1% |
| Chemical compound, drug | Alexa Fluor 546 Phalloidin | Thermo Fisher | Cat # A22283 | IF: 1:100 |
| Software, algorithm | GraphPad Prism 9.4.0 | GraphPad Software Inc | RRID:SCR_002798 | https://www.graphpad.com/ |
| Software, algorithm | Fiji (Image J) | *Schmidt et al., 2012* | RRID:SCR_002285 | http://imagej.net/Fiji |

## Plasmids and mutagenesis

For information on all the plasmids used in this study, see the Key Resources Table. Site-directed mutagenesis was performed by overlap extension PCR. Amplifications for all cloning were performed with Platinum SuperFi DNA polymerase (Thermofisher), and all finished constructs were validated by DNA sequencing (Eurofins Genomics).

## Cell culture and transfections

All cell lines were grown at 37 °C and 5% $CO_2$ in a humidified atmosphere and were regularly tested to ensure they were mycoplasma-free. hTERT-RPE1 cells were cultured in DMEM/F12 basal medium supplemented with 10% fetal bovine serum (FBS) and were reverse transfected using JetPrime (Polyplus-transfection), and their cilia analyzed 48 hr later, after 24 hr of serum starvation. HEK293T cells were maintained in DMEM +10% FBS, transfected using the calcium phosphate method, and lysed 40–48 hr later. HeLa cells were cultured in DMEM +10% FBS +Penicillin/Streptomycin and transfected using FuGENE 6 (Promega). Co-recruitment assays were performed 24 hr after transfection. CRISPR-engineered control and *Csnk2a1*-null mouse embryonic fibroblasts have been described elsewhere (*Loukil et al., 2021*). Raji B and Jurkat T (clone JE6.1) cell lines from ATCC were cultured in RPMI-1640 medium containing L-glutamine, penicillin/streptomycin, and 10% heat-inactivated FBS. All cell lines used in this study have been validated and used extensively in previous publications, with the exception of CRISPR cell lines, whose validation is described below.

## Immunofluorescence microscopy

For information on all the antibodies used in this study, see the Key Resources Table. hTERT-RPE1 cells grown to confluence on coverslips were fixed 5 min in PBS +4% paraformaldehyde (PFA) at room temperature (RT), followed by freezer-cold methanol for 3 min at −20 °C. Cells were then blocked and permeabilized for 30–60 min at RT in PBS +0.1% Triton X100 +2% donkey serum +0.02% sodium azide (blocking solution). Coverslips were then incubated in a humidified chamber for 2 hr at RT (or overnight, 4 °C) with blocking solution-diluted primary antibodies. After three PBS washes, PBS-diluted secondary antibodies and DAPI (1 µg/ml, Thermofisher) were added for 1 hr at RT in the dark. After three more PBS washes, coverslips were mounted on slides using Prolong Diamond (Thermofisher),

incubated overnight at 4 °C, and imaged with a Nikon Ti fluorescence microscope. Brightness and contrast of microscopic images were adjusted for optimal visualization using Adobe Photoshop or Fiji (Image J). For methanol-only fixation (*Figure 4—figure supplement 3*), cells were fixed 5 min in freezer-cold methanol at –20 °C, washed thrice in PBS, blocked as above but without Triton X100, and stained with antibodies as above.

## Generation of CRISPR cell lines

Puromycin-sensitive PAC-KO hTERT-RPE1 cells, and from them INPP5E-KO cells, were generated using CRISPR/Cas9 methods essentially as we described previously (*Barbeito et al., 2021*). The sequences of guide RNAs and information on the plasmids encoding them are in the Key Resources Table. PAC-KO cells were not analyzed genomically, but their puromycin sensitivity was confirmed in dose-response assays as shown in *Figure 4—figure supplement 1b-c*. The genomic analysis of INPP5E-KO cells was performed as previously reported (*Figure 4—figure supplement 2a*; *Barbeito et al., 2021*).

## TULP3 rescue assays in INPP5E-KO cells

For the rescue assays, INPP5E-KO hTERT-RPE1 cells were reverse transfected on coverslips using JetPrime (500 ng DNA and 1 µl JetPrime reagent, in 50 µl JetPrime buffer, were prepared following manufacturer's instructions and added to 2·10$^5$ suspended cells in 1 ml serum-containing medium, per well). Transfection medium was replaced 4 hr post-transfection (hpt). At 24 hpt, cells were starved with DMEM:F12 +0.2% FBS. At 48 hpt, cells were fixed (PFA +methanol), stained and visualized as described above. For quantitation, transfected-cell cilia without visible TULP3 staining were expressed as percentage relative to total transfected-cell cilia.

## Immunoprecipitation and western blot

HEK293T cells were lysed 40–48 hr post-transfection in buffer containing 50 mM Tris-HCl pH = 7.5, 150 mM NaCl, 1% Igepal CA-630 (Sigma-Aldrich) and 1 X Halt protease inhibitor cocktail (Thermofisher, #78429). Lysates were then rotated (15 min, 4 °C) and centrifuged (10 min, 20,000 g, 4 °C), and protein levels in the postnuclear supernatants measured with Pierce BCA Protein Assay Kit (Thermofisher). After equalizing protein amount and concentration in all samples, EGFP fusion proteins were immunoprecipitated with GFP-Trap magnetic agarose (GFP-Trap_MA, Chromotek) beads for 2 hr or overnight at 4 °C with rotation. Beads were then washed thrice in lysis buffer without protease inhibitors, eluted with 2 X Laemmli buffer containing 200 mM DTT, and boiled 5 min at 95 °C. SDS-PAGE and Western blots were performed as previously described (*Barbeito et al., 2021*; *Martin-Hurtado et al., 2019*). For information on all antibodies used for western blot, see the Key Resources Table.

## Phosphoinositide phosphatase assays

Activity assays were performed essentially as described (*Bielas et al., 2009*). Briefly, HEK293T cell lysates were obtained as above, and their protein levels measured with Pierce BCA Protein Assay kit (Thermofisher). EGFP-INPP5E or its mutants were then immunoprecipitated from 1 mg of cell lysate by overnight rotation at 4 °C with GFP-Trap_MA beads (Chromotek). Beads were then washed thrice in buffer containing 50 mM Tris-HCl pH = 7.5 and 150 mM NaCl buffer (no protease inhibitors or detergent), and twice more in activity buffer (50 mM Tris-HCl pH = 7.5, 150 mM NaCl, 3 mM MgCl$_2$ and 0.1% octyl-β-D-glucopyranoside (Alfa Aesar)). For the activity assays, beads were incubated in activity buffer supplemented with 120 µM diC8-PtdIns(4,5)P2, from Echelon Biosciences. After incubating the enzyme reactions for 20 min at 37 °C, the supernatant was retrieved from the beads and its phosphate concentration measured at 620 nm using the Malachite Green Assay Kit (Echelon Biosciences). Beads were then processed for western blot as above.

## Co-recruitment assays

Co-recruitment assays were performed using Eclipse Ti microscope (Nikon, Japan) with a 100 X TIRF objective (1.0 X zoom and 4X4 binning) in TIRF mode and PCO-Edge 4.2 BI sCMOS camera (PCO, Germany), driven by NIS Elements software (Nikon) and equipped with 440 nm, 514 nm, and 561 nm laser lines. Time lapse imaging was performed at 2 min intervals for 20 min, with 100 nM

rapamycin addition after the fifth time point. All live cell imaging was conducted at 37 °C, 5% $CO_2$ and 90% humidity with a stage top incubation system (Tokai Hit). Vitamin and phenol red-free media (US Biological) supplemented with 2% FBS were used in imaging to reduce background and photo-bleaching. Adequate co-expression of all relevant plasmids was confirmed by fluorescent imaging at appropriate wavelengths. To minimize variability due to relative expression levels, only cells showing at least 30% increase in mCherry intensity after addition of rapamycin were considered for quantification. All image processing and analyses for co-recruitment assays were performed using Metamorph (Molecular Devices, Sunnyvale, CA, USA) and FIJI software (NIH, Bethesda, MD, USA). Co-recruitment assay graphs show means ±95% confidence interval with n≥40 different cells pooled from at least three independent experiments. For information on plasmids used in these assays, see the Key Resources Table.

### Immune synapse analyses

Raji cells were attached to glass-bottom microwell culture dishes (IBIDI) using poly-L-lysine (20 µg/mL). Raji cells were then labeled with 10 µM CMAC (7-amino-4-chloromethylcoumarin, Molecular Probes), pulsed with 1 µg/ml SEE (Staphylococcus enterotoxin E, Toxin Technologies), and mixed with Jurkat T cells (clone JE6.1) (ATCC). To promote synaptic conjugate formation, cell-containing dishes were centrifuged at low speed (200xg, 30 seg) and incubated 5 min at 37 °C. For recombinant protein expression, exponentially growing Jurkat T cells were electroporated with 20–30 µg of plasmid as previously reported (*Herranz et al., 2019*), and Raji cells were added as above 40–48 hr post-electroporation. Cell conjugates were then fixed, first with PBS +2% PFA (10 min, RT), then with cold acetone (10 min at –20 °C). Immunofluorescence staining was done as previously described (*Herranz et al., 2019*). Imaging was performed using a Nikon Eclipse TiE microscope equipped with a DS-Qi1MC digital camera, a PlanApo VC 60 x NA 1.4 objective, and NIS-AR software (all from Nikon). Epifluorescence images were then deconvolved with Huygens Deconvolution Software from Scientific Volume Image (SVI), using the 'widefield' optical option.

### Structural analyses

Swiss PDB Viewer was used to visualize the crystallographic structure of INPP5E aa 282–623 (Protein Data Bank (PDB) ID: 2xsw) in order to identify candidate CLS or ciliopathy residues near the FDRxLYL motif (*Tresaugues et al., 2010*). Final figures were rendered using the PyMOL Molecular Graphics System (Version 2.0 Schrödinger), from either the PDB structure or the full length AlphaFold model (AF-Q9NRR6-F1) (*Jumper et al., 2021*).

### Quantitations and statistical analyses

For quantitation of Western blot levels, including co-IP experiments, Fiji/Image J was used. Using 8-bit blot images, average pixel intensity (API) and surface area of a rectangular region of interest (ROI) containing the protein band or background region was obtained using the *Measure* function. Data were then transferred to an Excel file, where API values were inverted (so that blacker in blot means higher intensity: API'=255 API), background-subtracted, and the resulting specific intensity multiplied by the corresponding ROI's surface area to get the band's total specific intensity. Ratios were then calculated for each sample as appropriate (e.g. EGFP-INPP5E construct levels normalized to tubulin, or co-IPed protein normalized to IPed protein, and to its own levels in lysates). Specific details for each experiment are described in figure legends. Quantitation of intensities along cilia length in the methanol immunofluorescence stainings were obtained using Fiji/Image J's *Plot Profile* function. For graphs and statistical analyses, GraphPad Prism 9 software was used. Details of statistical analyses are found in the corresponding figure legends.

## Acknowledgements

We thank Dr. Jung-Chi Liao for advice with INPP5E immune synapse stainings. This publication is part of grants PID2019-104941RB-I00 (to FRGG) and PID2020-114148RB-I00 (to MI), both funded by MCIN/AEI/ 10.13039/501100011033, and the latter also funded by ERDF, a way of making Europe. FRGG is also recipient of an ACCI-2020 grant from CIBERER. This work was also funded by NIH grant R01HD099784 (SCG), discretionary funds (TI), American Heart Association fellowship

20POST35220046 (ADR), Community of Madrid contract PEJD-2016/BMD-2341 (BSR), and MICINN predoctoral grant BES2016-077828 (RMM).

## Additional information

### Competing interests

Sarah C Goetz: has consulted for Arvinas Inc. The other authors declare that no competing interests exist.

### Funding

| Funder | Grant reference number | Author |
|---|---|---|
| Ministerio de Ciencia e Innovación | PID2019-104941RB-I00 | Raquel Martin-Morales Pablo Barbeito Francesc R Garcia-Gonzalo |
| Ministerio de Ciencia e Innovación | PID2020-114148RB-I00 | Manuel Izquierdo |
| Instituto de Salud Carlos III | CIBERER-ACCI-2020 | Francesc R Garcia-Gonzalo |
| National Institutes of Health | R01HD099784 | Sarah C Goetz |
| American Heart Association | 20POST35220046 | Abhijit Deb Roy |
| Comunidad de Madrid | PEJD-2016/BMD-2341 | Belen Sierra-Rodero |
| Ministerio de Ciencia e Innovación | BES2016-077828 | Raquel Martin-Morales |

The funders had no role in study design, data collection and interpretation, or the decision to submit the work for publication.

### Author contributions

Dario Cilleros-Rodriguez, Raquel Martin-Morales, Abhijit Deb Roy, Conceptualization, Formal analysis, Investigation, Methodology, Writing – review and editing; Pablo Barbeito, Formal analysis, Investigation, Methodology, Writing – review and editing; Abdelhalim Loukil, Conceptualization, Formal analysis, Investigation, Writing – review and editing; Belen Sierra-Rodero, Investigation, Writing – review and editing; Gonzalo Herranz, Investigation; Olatz Pampliega, Conceptualization, Resources, Writing – review and editing; Modesto Redrejo-Rodriguez, Data curation, Visualization, Writing – review and editing; Sarah C Goetz, Manuel Izquierdo, Takanari Inoue, Conceptualization, Supervision, Funding acquisition, Writing – review and editing; Francesc R Garcia-Gonzalo, Conceptualization, Formal analysis, Supervision, Funding acquisition, Writing – original draft, Project administration, Writing – review and editing

### Author ORCIDs

Dario Cilleros-Rodriguez http://orcid.org/0000-0001-9529-5320
Raquel Martin-Morales http://orcid.org/0000-0003-0933-6160
Pablo Barbeito http://orcid.org/0000-0003-0758-0012
Abhijit Deb Roy http://orcid.org/0000-0003-1640-2402
Olatz Pampliega http://orcid.org/0000-0002-7924-6374
Sarah C Goetz http://orcid.org/0000-0001-9705-6390
Manuel Izquierdo http://orcid.org/0000-0002-7701-1002
Takanari Inoue http://orcid.org/0000-0002-7957-7624
Francesc R Garcia-Gonzalo http://orcid.org/0000-0002-9152-2191

### Decision letter and Author response

Decision letter https://doi.org/10.7554/eLife.78383.sa1
Author response https://doi.org/10.7554/eLife.78383.sa2

## Additional files

### Supplementary files
• Transparent reporting form

### Data availability
All data generated during this study are included in the manuscript and associated files. Source data files are provided for Figures 2,3,4,5,6,7,8, and for Figure Supplements 1-1, 2-2, 2-3, 2-4, 3-1, 3-2, 7-1.

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
