## [Editor Report]

This manuscript is of interest to readers in the field of cilia biology and ciliopathies. The authors address the molecular mechanisms by which INPP5E, a ciliary phosphoinositide phosphatase mutated in multiple ciliopathies, is targeted to the primary cilium of cultured mammalian cells. By combining cell-based analysis of various INPP5E mutant constructs with biochemical assays and structure prediction, the authors show that ciliary targeting of INPP5E requires its folded catalytic domain and is controlled by four ciliary localization signals. The work clarifies and extends previous work in the field and reveals a complex ciliary targeting mechanism unparalleled among other known ciliary cargoes. The claims are generally well supported by the data.

---

## [Decision Letter]

**Decision letter after peer review:**

Thank you for submitting your article "Multiple Ciliary Localization Signals Control INPP5E Ciliary Targeting" for consideration by *eLife*. Your article has been reviewed by 3 peer reviewers, including Lotte B Pedersen as Reviewing Editor and Reviewer #1, and the evaluation has been overseen by Piali Sengupta as the Senior Editor. The following individual involved in the review of your submission has agreed to reveal their identity: Esben Lorentzen (Reviewer #3).

Essential revisions:

1) The functional significance of the identified CLSs should be addressed in more detail (for example, examination of the role the CLSs play in cilia protein or phosphoinositide composition, cilia-dependent signaling pathways, cilia assembly or disassembly), which would provide stronger evidence of their biological significance and provide a conceptual advance in the cilia field essential to development and disease. This is particularly important considering that INPP5E mutations associated with JBTS have variable effects on INPP5E cilia localization.

2) Indirect immunofluorescence analysis in fixed cells has been used extensively here to demonstrate the identification and characterization of cilia localization signals in INPP5E. A recent publication in this field has reported possible artifactual INPP5E localization issues dependent upon fixation conditions (doi:10.3389/fcell.2021.634649). The authors should assess their data in the context of this new report and repeat a part of their principal localization experiments with alternative fixation conditions to address any possible artifactual issues with INPP5E ciliary localization. This would increase confidence in the co-localization studies.

3) For all IFM figures comparing cilia localization of various EGFP- INPP5E mutants, the authors should provide information about how many times experiments were repeated, and how many cells/cilia were analyzed in each case. In Figures 2d and 3c, the authors do provide a quantitative analysis of cilia localization of selected mutants, but the number of cells examined is not stated in the legend, nor in the Methods section. Furthermore, it is unclear whether relative expression levels and/or stability of the different INPP5E mutant constructs in RPE1 cells are similar? This is important to clarify as it could influence their cilia localization.

4) The claim that the authors have elucidated the mechanisms of INPP5E ciliary targeting requires further evidence that has not been addressed in the immunoprecipitation studies described within the manuscript. The authors use the terminology "decreased binding" when actually referring to "decreased immunoprecipitation". This language needs to be clarified in the text as a reduction in co-immunoprecipitation could also be due to protein misfolding in the mutants. Furthermore, it would be desirable if authors could somehow quantify the IP results, e.g. by measuring band intensities and quantifying ARL13B levels in each IP pellet relative to the amount of EGFP- INPP5E levels in the same IP pellets. Before a claim of "decreased binding" can be substantiated, the authors also need to validate that the mutant proteins are correctly folding and confirm immunoprecipitation data with direct binding studies; alternatively, this claim needs to be adjusted/toned down.

---

## [Author Response]

Essential revisions:1) The functional significance of the identified CLSs should be addressed in more detail (for example, examination of the role the CLSs play in cilia protein or phosphoinositide composition, cilia-dependent signaling pathways, cilia assembly or disassembly), which would provide stronger evidence of their biological significance and provide a conceptual advance in the cilia field essential to development and disease. This is particularly important considering that INPP5E mutations associated with JBTS have variable effects on INPP5E cilia localization.

To address this great point, we have assessed CLS function in a rescue assay of TULP3 ciliary accumulation in INPP5E-KO cells (Figure 4c-d). This assay is based on the known fact that INPP5E-KO cilia have abnormally high levels of TULP3, as we and others showed some years ago (refs. 33-34). The results and quantitation of these rescue experiments show that all CLS mutants failing to localize to cilia also have significantly reduced activity in this rescue assay, as compared to INPP5E WT (Figure 4c-d). Therefore, these CLSs are indeed important regulators of INPP5E ciliary function.

To carry out these assays, we have used INPP5E-KO hTERT-RPE1 cells that we have generated using CRISPR-Cas9 technology. Generation of these cells is described in Figure 4—figure supplement 2. These INPP5E-KO cells were in turn generated from puromycin-sensitive hTERT-RPE1 cells, which we previously generated by inactivating, also via CRISPR, the puromycin acetyltransferase (PAC) gene in the original hTERT-RPE1 cell line (which contains this gene as a remnant of their immortalization procedure). The generation of these PAC-KO hTERT-RPE1 cells, which can be a useful resource for other labs, is described in Figure 4—figure supplement 1.

The manuscript text has been updated to incorporate all these changes. In addition to the cited figures and their legends, the changes can be found in Results (lines 326-353), Discussion (lines 594-595), and Methods (lines 731-745).

2) Indirect immunofluorescence analysis in fixed cells has been used extensively here to demonstrate the identification and characterization of cilia localization signals in INPP5E. A recent publication in this field has reported possible artifactual INPP5E localization issues dependent upon fixation conditions (doi:10.3389/fcell.2021.634649). The authors should assess their data in the context of this new report and repeat a part of their principal localization experiments with alternative fixation conditions to address any possible artifactual issues with INPP5E ciliary localization. This would increase confidence in the co-localization studies.

As suggested, we have repeated our key INPP5E localization experiments using alternative fixation conditions. Specifically, we have looked at all CLS mutants with methanol fixation. In agreement with the above-mentioned paper, in methanol-fixed hTERT-RPE1 cells, we often see EGFP-INPP5E WT accumulated at the ciliary base, with some staining also seen along the axoneme. In contrast, CLS mutants are never seen inside the axoneme under these conditions, confirming that these CLSs grant INPP5E access to the axonemal compartment (Figure 4—figure supplement 3a-b).

Three of the five CLS mutants (∆CLS1, ∆CLS4 and ∆CLS1+4) strongly accumulate at the transition zone (TZ) under these fixation conditions (Figure 4—figure supplement 3c), indicating that these CLSs play a role in INPP5E’s TZ crossing. This is no surprise for CLS4 (CaaX box), which was already known to function at the TZ (as explained in lines 83-85 of the introduction). It also makes sense for CLS1, as we show its function is partially redundant with CLS4 (Figure 3a-c). Instead, ∆CLS3 is never seen at the ciliary base in methanol-fixed cells, and ∆CLS2 only occasionally so. This suggests a different mechanism of action for these CLSs, perhaps in retaining INPP5E inside cilia, or in reaching the ciliary base.

Regarding possible artefacts, it seems clear (from the aforementioned Conduit et al., 2021 paper, and from our own data) that the different ciliary pools of INPP5E are better seen with different fixations. Thus, as the above paper shows, the ciliary base accumulation of EGFP-INPP5E seen in live hTERT-RPE1 cells is largely lost with paraformaldehyde (PFA) fixation. On the other hand, our data with ∆CLS1 and ∆CLS4 suggest that methanol fixation is worse at preserving axonemal staining (since we see these mutants inside the axoneme with PFA (Figure 3b-c), but not methanol (Figure 4—figure supplement 3a-b)). In summary, we think that each fixation method has its own pros and cons when it comes to visualizing INPP5E.

In addition to Figure 4—figure supplement 3 and its legend, associated changes in the manuscript can be found in Results (lines 354-373), Discussion (lines 625-629), and Methods (lines 727-730 and 814-816). Thanks for raising this excellent point.

3) For all IFM figures comparing cilia localization of various EGFP- INPP5E mutants, the authors should provide information about how many times experiments were repeated, and how many cells/cilia were analyzed in each case. In Figures 2d and 3c, the authors do provide a quantitative analysis of cilia localization of selected mutants, but the number of cells examined is not stated in the legend, nor in the Methods section. Furthermore, it is unclear whether relative expression levels and/or stability of the different INPP5E mutant constructs in RPE1 cells are similar? This is important to clarify as it could influence their cilia localization.

We have added to the figure legends all the information regarding number of independent experiments and of cells and cilia analyzed.

We have also analyzed protein levels of the EGFP-INPP5E mutants we use throughout the paper. To do this, we have performed Western blots of transfected HEK293T cells, and quantitated the levels of each EGFP-INPP5E protein relative to tubulin expression. These data are found in Figure 1—figure supplement 1, Figure 2—figure supplement 2, Figure 3—figure supplement 2, and Figure 5d.

In the text, these data have been incorporated to Results (lines 163-181, 220-224, 290-293, 304-305 and 383-399), Discussion (lines 581-595), and Methods (lines 755-756 and 806-814) sections.

Thanks for these great points, which have clearly improved the quality of our data.

4) The claim that the authors have elucidated the mechanisms of INPP5E ciliary targeting requires further evidence that has not been addressed in the immunoprecipitation studies described within the manuscript. The authors use the terminology "decreased binding" when actually referring to "decreased immunoprecipitation". This language needs to be clarified in the text as a reduction in co-immunoprecipitation could also be due to protein misfolding in the mutants. Furthermore, it would be desirable if authors could somehow quantify the IP results, e.g. by measuring band intensities and quantifying ARL13B levels in each IP pellet relative to the amount of EGFP- INPP5E levels in the same IP pellets. Before a claim of "decreased binding" can be substantiated, the authors also need to validate that the mutant proteins are correctly folding and confirm immunoprecipitation data with direct binding studies; alternatively, this claim needs to be adjusted/toned down.

We agree that the claim in the abstract that “*we elucidate the mechanisms*” of INPP5E ciliary targeting sounded excessive. We have replaced it by “*we shed light on the mechanisms*”, which we believe it indicates simply that we know more now than we did before. This sentence not only refers to the co-IPs, but also to our identification of novel CLSs.

Regarding our claims of “*decreased binding*” (or similar expressions), we have replaced many of them by “*decreased co-IP*” (or equivalent). That said, when we speak of *binding*, *interaction* or *association* in the context of our co-IPs, we do not intend to make any mechanistic presumptions as to how the interaction occurs (e.g. whether direct or not). Therefore, we see no need to prove that an interaction is direct in order to speak in terms of *decreased binding* (but of course, this depends on how one defines the word *binding*).

Moreover, although we agree in principle that lack of co-IP may be due to misfolding, we do not think this is a major issue in our data. For CLS1 and CLS4, we show that their mutants are fully enzymatically active (Figure 3—figure supplement 1), and must hence be properly folded. For CLS2-3, although it is true that their mutation reduces enzyme activity, they still preserve significant activity (Figure 2e), their protein levels and stability are not strongly reduced (Figure 2—figure supplements 2-3), and they co-IP with all proteins analyzed except TULP3 (Figures 6-8). Therefore, on this count we also think it is reasonable for us to speak in terms of binding/interaction/association, as we do sometimes.

Still, as mentioned above, we have replaced these terms in multiple places for “co-IP” (as a result, searching the manuscript for “co-IP” now retrieves 50 hits, as opposed to 23 in the previous version).

Lastly, we have quantified all co-IP results throughout the paper. We have done it as suggested by the reviewer: normalizing co-IPed protein by amount of EGFP-INPP5E IPed in the same sample (and by lysate levels of the co-IPed protein). These quantitations have been incorporated into Figure 6a (PDE6D), Figure 6b (RPGR), Figure 6d (ARL13B), Figure 7b-c (TULP3), Figure 7f (CEP164), and Figure 8b (ATG16L1). Details on each quantitation are found in the figure legends, and in Methods (lines 806-814). The corresponding Results subsections (lines 456-466, 481-482, 505-506 and 553-557), and the Discussion (lines 617-620), have been updated accordingly.

Thanks for the great point. We think it has greatly improved the quality of our work.